# Effects of Combined Periodontal, Endodontic, and Dentoalveolar Surgical Treatments on Laboratory Parameters in Patients with Hyperlipidemia—A Clinical Interventional Study

**DOI:** 10.3390/jcm14010241

**Published:** 2025-01-03

**Authors:** Renáta Martos, Mariann Harangi, Judit Szabó, Anett Földvári, János Sándor, Éva Katona, Ildikó Tar, György Paragh, Csongor Kiss, Ildikó Márton

**Affiliations:** 1Department of Operative Dentistry and Endodontics, Faculty of Dentistry, University of Debrecen, Nagyerdei krt. 98, 4032 Debrecen, Hungary; 2Division of Metabolism, Department of Internal Medicine, Faculty of Medicine, University of Debrecen, 4032 Debrecen, Hungary; harangi@belklinika.com (M.H.); paragh@belklinika.com (G.P.); 3Institute of Health Studies, Faculty of Health Sciences, University of Debrecen, 4032 Debrecen, Hungary; 4Petz Aladár County Teaching Hospital, Microbiol. Lab. 9024. Győr. Vasvári Pál u. 2-4, 9024 Hungary, Hungary; szabjud77@gmail.com; 5Doctoral School of Health Sciences, University of Debrecen, 4032 Debrecen, Hungary; anettefoldvari@gmail.com; 6Department of Public Health and Epidemiology, Faculty of Medicine, University of Debrecen, 4028 Debrecen, Hungary; janos.sandor@med.unideb.hu; 7HUN-REN-DE Public Health Research Group, Department of Public Health and Epidemiology, Faculty of Medicine, University of Debrecen, 4028 Debrecen, Hungary; 8Division of Clinical Laboratory Science, Department of Laboratory Medicine, Faculty of Medicine, University of Debrecen, 4032 Debrecen, Hungary; ekatona@med.unideb.hu; 9Department of Oral Medicine, Faculty of Dentistry, University of Debrecen, 4032 Debrecen, Hungary; tar.ildiko@dental.unideb.hu; 10Division of Pediatric Hematology-Oncology, Department of Pediatrics, Faculty of Medicine, University of Debrecen, Nagyerdei krt. 98, 4032 Debrecen, Hungary; kisscs@med.unideb.hu; 11Department of Biochemistry and Molecular Biology, Faculty of Medicine, University of Debrecen, Nagyerdei krt. 98, 4032 Debrecen, Hungary; marton.ildiko@dental.unideb.hu; 12Faculty of Health Sciences, University of Debrecen, Kassai út 26, 4028 Debrecen, Hungary

**Keywords:** hyperlipidemia, C-reactive protein, non-surgical dental treatment, surgical dental treatment, cardiovascular disease

## Abstract

**Background:** Patients with hyperlipidemia are of interest because of the possible interplay between chronic local dental infections and hyperlipidemia. This interventional clinical study aimed to evaluate the oral health status of hyperlipidemic patients receiving lipid-lowering therapy for at least 6 months and the effects of non-surgical and surgical dental treatments on serum C-reactive protein (CRP) levels and lipid markers. **Methods:** Twenty-eight patients with controlled hyperlipidemia and 18 healthy controls were enrolled in the study. All participants underwent dental examinations (clinical evaluation, X-ray imaging, and microbial analysis of subgingival and supragingival plaque samples) at baseline. Hyperlipidemic patients received periodontal, endodontic, and dentoalveolar surgical treatments. Serum CRP and lipid parameters were assessed at baseline, 1 week, and 3 months, while subgingival and supragingival plaque samples were analyzed at baseline and 3 months after completing dental treatments. **Results:** At the 3-month follow-up, clinical periodontal characteristics, including the plaque index, gingival index, and periodontal probing depth, improved significantly (*p* < 0.05). A significant shift in microflora was observed in both subgingival and supragingival plaque samples (*p* < 0.05), alongside improvements in periodontal values and a significant reduction in serum CRP levels (*p* < 0.05). Serum cholesterol levels decreased significantly, while moderate improvements in serum triglycerides, low-density lipoprotein, and high-density lipoprotein levels were observed but were not statistically significant (*p* > 0.05). **Conclusions:** Treating local dental inflammation is associated with a significant decrease in CRP and cholesterol levels and may serve as beneficial adjunct therapy alongside lipid-lowering therapy in patients with hyperlipidemia.

## 1. Introduction

The possible interplay between chronic local inflammation and systemic diseases has been highlighted by review articles summarizing longitudinal observational studies, cross-sectional studies, and case–control studies [1,2,3,4]. Cardiovascular disease (CVD) is a leading cause of mortality worldwide each year [5], and it has been associated with dental infections [6,7]. Increasing attention has been drawn to the observation that a higher frequency of dental infections, such as endodontic infections, gingivitis, and periodontitis; a higher number of missing teeth; and poorer oral health have been identified in patients with chronic heart disease, acute myocardial infarction, or CVD across various populations compared with healthy control subjects [1,8,9,10,11,12,13,14,15]. Furthermore, it is worth noting that among health-conscious patients and those of higher socioeconomic status, root canal treatment (RCT) is often preferred over extraction. Therefore, in addition to missing teeth, the number of RCTs should also be considered when assessing patients’ risk of oral inflammation-related CVD [7,11].

Hyperlipidemia is a well-known risk factor for future CVD events [16], but the periodontal condition can also influence the risk of CVD [3,17,18,19,20,21,22,23,24,25,26]. The systemic effects of plaque-induced periodontal disease and its treatment on the serum C-reactive protein (CRP) level, as well as the potential association between severe periodontal disease and hyperlipidemia, have been previously described [1,2,15,17,18,19,24,26,27,28,29,30]. Several studies have revealed that poor oral hygiene and hyperlipidemia often co-exist, ultimately leading to an increased risk of developing CVD [15,17,19,26]. Consequently, multiple studies conducted over the last decade have evaluated the periodontal condition of hyperlipidemic patients [17,19,20,21,22,29] and the efficacy of various periodontal therapies in hyperlipidemic patients with periodontitis [19,20,21,22].

Examination of the possible interplay between apical periodontitis and systemic diseases has evidenced a moderate risk for CVD, although this relationship is influenced by several confounding factors. Predicting the effects of endodontic treatment in isolation remains challenging [31,32]. The local and systemic immunological effects of chronic apical periodontitis [33,34] and the impact of combined RCT and surgical treatment on serum CRP levels in populations with good general health have been previously reported [35,36]. The systemic effects may result from coexisting pulpal and chronic apical inflammation, triggered by anaerobic bacteria. This process, similar to plaque-induced periodontal disease, is associated with cytokine production and tissue destruction [37,38,39]. Furthermore, low-density lipoprotein (LDL) may interact with Toll-like receptor 4, similarly to how lipopolysaccharide (LPS) from the bacterial walls of periodontopathogenic bacteria functions, while Toll-like receptor 2 can also interact with LPS from Gram-positive bacteria and bind LDL [40]. This could represent an important pathway linking periodontal disease and hyperlipidemia.

The systemic effects of concurrently treating pulpal and periapical inflammation in hyperlipidemic patients have not been previously evaluated. Therefore, our interventional clinical study aimed to assess the impact of combined periodontal, endodontic, and dentoalveolar surgical treatments on serum CRP levels and serum lipid components and lipoproteins in patients with hyperlipidemia. To the best of our knowledge, evidence of the effects of these treatment modalities on laboratory parameters in hyperlipidemic patient groups is lacking. We hypothesized that in addition to the beneficial effects of treating periodontal disease, eliminating microbial sources associated with pulpal inflammation could provide adjunctive therapeutic support for hyperlipidemic patients.

## 2. Materials and Methods

### 2.1. Study Design

The study was a collaboration between the 1st Department of Internal Medicine, Medical Health and Science Center, University of Debrecen (MHSC UD), and the Department of Restorative Dentistry, Faculty of Dentistry, MHSC UD. It was conducted in compliance with the requirements of the Ethical Committee of the MHSC UD (Registration number: 2177-2004 DEOEC RKEB 05.04.2004) and the Code of Ethics of the World Medical Association. The study adhered to the ethical standards outlined in the 1964 Declaration of Helsinki, as revised in 2000. All participants provided written informed consent to participate. The study design is illustrated in Figure 1.

### 2.2. Participants

#### 2.2.1. Hyperlipidemic Patient Group

Twenty-eight Caucasian patients with Fredrickson type IIa and IIb hyperlipidemia were enrolled in the study. Type IIa, also known as familial hypercholesterolemia, occurs due to a gene alteration and causes high levels of LDL cholesterol (LDL-C) (total cholesterol > 5.2 mmol/L, LDL-C > 3.0 mmol/L). Type IIb is a subvariation involving a gene alteration that causes both high cholesterol and high triglycerides (mixed hyperlipidemia) (total cholesterol > 5.2 mmol/L, LDL-C > 3.0 mmol/L, triglycerides > 1.7 mmol/L) [41]. Lipid-lowering therapy included simvastatin (20–40 mg/day), atorvastatin (20–40 mg/day), fluvastatin (80 mg/day), or rosuvastatin (10–40 mg/day) as monotherapy or in combination with ezetimibe (10 mg/day) and/or fenofibrate (160 mg/day), administered according to current therapeutic guidelines. Patients with hyperlipidemia were selected from those attending regular medical examinations at the 1st Department of Internal Medicine, MHSC UD. The exclusion criteria were alcoholism, liver disease, elevated liver enzymes, recent myocardial infarction, autoimmune and infectious diseases, pregnancy and lactation, edentulism, oral mucosal disease, and any other dental or periodontal disease requiring antibiotic treatment in the previous 6 months. Smokers were also excluded. The lipid treatment regimens were kept constant throughout the study. To avoid the short-term effects of lipid-lowering therapy, patients were enrolled after the adjustment of combined dietary and statin/fibrate therapy for at least 6 months. This ensured that the confounding effects of these agents on lipid and inflammatory parameters were excluded. According to matching guidelines, some patients received other medications, including antihypertensive agents and platelet aggregation inhibitors, at constant doses. All comorbid conditions were under strict and continuous control by specialists.

#### 2.2.2. Control Group

Age- and sex-matched healthy controls willing to participate in this study were selected from patients attending check-ups at the Department of Restorative Dentistry, Faculty of Dentistry, MHSC UD. Controls were selected to exclude individuals with hyperlipidemia or hypertension, and smokers were also excluded. Body mass index was calculated by a physician.

### 2.3. Dental Examinations

Participants in both the test and control groups underwent clinical and radiological examinations. Orthopantomograms and radiovisiography (Planmeca PM 2002 CC PROLINE, Helsinki, Finland) were used to evaluate root canal obturations and detect chronic apical lesions or other radiolucencies.

Oral examinations were conducted by the same dentist. The number of teeth, type of restorations, caries, endodontically treated teeth, roots, residual roots, impacted teeth, tooth loss, and periapical radiolucencies were recorded, excluding third molars. Teeth with doubtful vitality were tested using an electrical vitality tester (Pulppen B1000 Analogous Dental Electronic, Ballerup, Denmark). Periodontal examinations included the mesial, distal, buccal, and palatal/lingual aspects of all teeth, excluding the third molars. For characterization of the periodontal condition, the periodontal probing depth (PPD) was measured from the gingival margin to the base of the clinical pocket using a UNC probe 15 (HU-Friedy, Milano, Italy) [42,43]. The plaque index (PI) (Sillness–Löe) [44] and gingival index (GI) (Löe–Sillness) [45] were also recorded. The PI, GI, and PPD were re-evaluated 3 months after the completion of complex dental treatments.

### 2.4. Reproducibility of Measurements

A single examiner (R.M.) conducted all assessments throughout the study. The calibration process involved repeated clinical periodontal examinations of five patients. Each patient was assessed twice during one visit, with a 1 h interval between assessments. The mean differences in the scores of examined sites were measured to an accuracy of ±1 mm or ±1 score. Reproducibility was considered acceptable with at least 85% accuracy. No significant differences were observed between the periodontal values (PI, GI, and PPD) from the two assessments.

### 2.5. Microbiological Examination

Microbial examinations were conducted using subgingival and supragingival plaque samples collected before and 3 months after complex dental treatments. Bacterial samples were taken from the site with the deepest probing depth, as identified by the same dentist during the initial examination, and the identical site was re-examined 3 months after the combined dental treatments.

Under cotton roll isolation, sterile paper points were inserted with tweezers into the apical extent of the deepest sites. The absorbent points were then removed with tweezers, placed into a test tube, and immediately transferred to a sterile transport medium (Stuart transport medium, Remel, Lenexa, KS, USA) using the swab. Supragingival plaque culture was collected with a special sterile transport swab (Stuart transport medium). The test tubes were transported to the Bacteriology Laboratory of the Department of Medical Microbiology, MHSC UD, where the samples were processed according to standard laboratory methods. The presence of bacterial species was detected non-quantitatively, indicating whether each species was present or absent. The distribution of pathogenic versus non-pathogenic subgingival and supragingival plaque flora was expressed as a proportional value of all detected species.

### 2.6. Laboratory Culturing Method

Under strict aseptic conditions, a calibrated (10 µL) platinum loop was used to streak the homogenously dispersed material from the test tubes onto two Petri plates containing Columbia agar supplemented with 5% sheep blood, hemin (5 mg/L), and 0.4 µL/mL vitamin K1 (bioMérieux, Lyon, France). One Petri plate was incubated under aerobic conditions, while the other was incubated anaerobically for 5 days in a Ruskinn Concept 400 anaerobic chamber (Labtech, Rotherham, UK).

### 2.7. Assessment of Microbial Growth

Microbial growth in the Petri plates was assessed after 5 days. Bacteria were identified based on Gram staining, colony morphology, biochemical tests (e.g., catalase, coagulase, oxidase), and the VITEK 2 automated identification system (bioMérieux, Lyon, France), following the manufacturer’s instructions. The VITEK 2 system operates with specific cards for the identification of Gram-negative, Gram-positive, and anaerobic bacteria. Bacterial or fungal colonies were suspended in a 0.45% sodium chloride medium, with suspension turbidity adjusted to 0.5 McFarland for Gram-negative bacteria, 1.0 McFarland for Gram-positive bacteria, and 3.0 McFarland for anaerobic bacteria. The inoculums were introduced into the corresponding VITEK 2 cards and incubated for 18 to 20 h. The identification time varied depending on bacterial metabolism and the card type.

### 2.8. Examination of Serum Markers and Hyperlipidemia Treatment Regimens

Baseline blood samples were collected from all participants, while 1-week and 3-month follow-up samples were obtained only from hyperlipidemic patients as part of the lipid-lowering therapy protocol. Blood samples were drawn via venipuncture between 8:00 a.m. and 12:00 p.m., following a 12 h fasting period. Serum levels of CRP, lipids, and lipoproteins were measured before (baseline) and at 1 week and 3 months after combined dental treatments at the Clinical Research Center and the Department of Laboratory Medicine, MHSC UD. CRP serum levels were quantified using commercial high-sensitivity immunoassay kits for human sera (Pharmingen BD OptEIA™ Human CRP ELISA Set, San Diego, CA, USA) according to the manufacturer’s instructions. Serum lipids were quantified using standard clinical pathology methods: serum cholesterol and triglyceride levels were measured using enzymatic colorimetric tests (GPO-PAP, Modular P-800 Analyzer; Roche/Hitachi, Basel, Switzerland), while high-density lipoprotein cholesterol (HDL-C) was assessed using a homogenous enzymatic colorimetric assay (Roche HDL-C plus 3rd generation, Mannheim, Germany). The LDL-C fraction was calculated indirectly using the Friedewald equation. Pathological lipid values were defined based on the laboratory’s recommended cut-off points: triglycerides > 1.7 mmol/L, cholesterol > 5.2 mmol/L, HDL-C < 1.3 mmol/L (women) or < 1.0 mmol/L (men), and LDL-C > 3.4 mmol/L.

### 2.9. Dental Treatments

The treatment modalities performed during the study are detailed in Figure 2. Hyperlipidemic patients received oral hygiene instructions, including motivation and guidance (clarifying periodontal issues and targeting plaque reduction) for tooth brushing and interdental cleaning. Scaling and root planing were performed, involving the removal of major deposits using ultrasound scaling, subgingival surface planing with hand scalers, and polishing. Caries control was addressed with fluoride-containing tooth gels. No antibiotic treatment was administered. All dental treatments were performed by the same dentist [46].

RCT and dentoalveolar surgical therapies, including extractions, apicectomies, and the removal of roots or impacted teeth, were carried out according to accepted professional protocols. The indications for the different treatment modalities are shown in Figure 2. Surgical periodontal therapy, filling therapy, and prosthetic rehabilitation were not included in the dental treatment modalities applied during the course of the study.

### 2.10. Outcomes

The primary outcomes were improvements in periodontal parameters and findings from plaque and sulcus samples at the 3-month follow-up. The secondary outcomes were improvements in serum CRP and lipid levels at the 3-month follow-up.

### 2.11. Statistical Analysis

The required sample size was calculated based on the standard deviation of the CRP levels (1.3 mmol/L) among patients without malignant or inflammatory diseases, as estimated by the Department of Laboratory Medicine, with a 1-mmol/L CRP concentration defined as the minimum detectable change. Age- and sex-matched healthy controls (n = 28; mean age, 52.11 ± 10.69 years; 10 men and 18 women) for CRP evaluation were selected from the control pool of the TECH-09-A1-2009-0113 (mAB-CHIC) project of the Department of Laboratory Medicine, MHSC UD. Assuming equal group sizes for controls and patients, a 5% type I error, and 80% statistical power, the required minimum sample size was 27 patients in the diseased group. The CRP control group data was used exclusively for sample size determination.

The measured parameters are presented as mean ± standard deviation for normally distributed data. Following normality testing, the results in the test and control groups were compared using independent-sample *t*-tests. Changes in the mean values of the patients’ baseline and 3-month results were analyzed with paired *t*-tests. Associations between the occurrence of pathogenic and non-pathogenic flora in dental and subgingival plaque samples were assessed using the chi-square test. Relationships between changes in dental parameters and detected serum values were analyzed with Spearman’s correlation, using the differences between the baseline and 3-month values. Potential causative effects were evaluated with repeated-measures analysis of variance.

## 3. Results

Based on the inclusion criteria, 38 hyperlipidemic patients were initially enrolled in the study. Six participants declined to participate, and four did not complete the treatment period because of non-compliance. Ultimately, 28 patients with hyperlipidemia (mean age, 53.36 ± 11.60 years; 11 men and 17 women) and 18 healthy controls (mean age, 51.22 ± 11.80 years; 6 men and 12 women) completed the study (Figure 1). During the study period, the participants reported no changes in lifestyle or medications, and no adverse events or side effects were observed.

### 3.1. Baseline Characteristics

There were no significant differences in age between patients and healthy controls (*p* = 0.548). At baseline, the mean number of missing teeth was similar between the study group (10.32 ± 6.56) and healthy controls (7.39 ± 6.66) (*p* = 0.148). Significant differences (*p* < 0.01) were observed between patients and healthy controls in the PI (1.02 ± 0.37 vs. 0.34 ± 0.20), GI (0.71 ± 0.52 vs. 0.25 ± 0.25), and PPD (2.02 ± 0.54 vs. 1.50 ± 0.25 mm) (Table 1). Additionally, a significant difference was detected in body mass index (30.23 ± 6.40 vs. 26.06 ± 4.24 kg/m^2^) (*p* = 0.019).

All 28 patients had chronic dental inflammation originating from pulpal or periodontal conditions. Of these, 5 patients were diagnosed with chronic gingivitis (according to EFP criteria, 2017 [42]), and 10 patients had a PPD of ≤3 mm. The remaining dental issues included pulp necrosis, chronic apical periodontitis, periapical cysts, insufficient RCT, radices, residual roots, and impacted teeth. Twelve patients had insufficient RCT, affecting 24 teeth, and in 9 cases, this was associated with chronic apical periodontitis. In three patients, chronic apical periodontitis was found in association with chronic pulpal inflammation, affecting three teeth. Four patients had residual roots (five roots in total), two of which were associated with chronic apical periodontitis. In one patient, a periapical cyst was identified and confirmed by postsurgical histology. The number of patients in each pathosis group is shown in Figure 3, while the number of teeth associated with the detected pathologies is displayed in Figure 4.

### 3.2. Periodontal Outcomes

Combined periodontal, endodontic, and dentoalveolar surgical treatments led to significant improvements in periodontal parameters 3 months after treatment. The PI decreased to 0.73 ± 0.47 (*p* < 0.01), the GI to 0.55 ± 0.39 (*p* < 0.05), and the PPD to 1.73 ± 0.42 mm (*p* < 0.01) (Table 1).

Significant differences were observed in the microbial analysis of subgingival and supragingival plaque samples obtained on admission between patients and healthy controls, as well as during the follow-up period within the patient group, with respect to the frequency of normal and pathogenic flora (*p* < 0.01). The detected species in dental plaque and sulcus samples are presented in Figure 5 and Figure 6. The occurrence of pathogenic flora had decreased significantly by the 3-month follow-up investigation in both subgingival and supragingival plaque samples (*p* < 0.001) (Table 2).

Subgingival plaque samples on admission contained *Prevotella* species, including *P. intermedia*; *Capnocytophaga* species; *Fusobacterium* species; and *Bacteroides* species, such as *B. eggerthii* and *B. caccae*. Other detected organisms included *Streptococcus intermedius*, Gram-negative rods, Gram-negative cocci, and Gram-positive cocci. Notably, periodontal pathogens of the red complex were not cultivated. By the 3-month follow-up, normal flora was predominant in most samples, although *Prevotella corporis* and Gram-positive bacteria were still occasionally detected.

Supragingival plaque samples on admission contained *Capnocytophaga* species, *Prevotella* species (e.g., *P. corporis*, *P. intermedia*, *P. oralis*, *P. disiens*, *P. loeschii*), *Bacteroides* species, *Fusobacterium*, Gram-negative cocci and rods, and Gram-positive cocci. By the 3-month follow-up, a marked reduction in periodontopathogenic bacteria was observed. *Capnocytophaga* species, *P. corporis*, *Enterobacter* species, and Gram-negative and Gram-positive bacteria were detected less frequently and in smaller amounts.

### 3.3. Serum CRP and Lipid Levels

No significant difference in the serum CRP levels was observed between hyperlipidemic patients and healthy controls at baseline (*p* = 0.14) (Table 3). However, 3 months after completing the periodontal, endodontic, and dentoalveolar surgical treatments, the serum CRP levels in the hyperlipidemic patient group decreased significantly from 3.30 ± 2.94 to 2.29 ± 1.73 mg/L (*p* < 0.05) (Table 3). Regarding lipid levels, the applied lipid-lowering therapy resulted in no significant differences in LDL-C or HDL-C levels (*p* > 0.05), while a borderline difference was noted in total cholesterol levels (*p* = 0.051) and a significant difference in triglyceride levels between healthy controls and hyperlipidemic patients at baseline (*p* < 0.05) (Table 3). A significant reduction in serum cholesterol levels was observed between the baseline and 3-month results (*p* < 0.05), with a borderline reduction in LDL-C (*p* = 0.057). No significant changes were found in triglyceride levels (*p* > 0.05), and HDL-C levels remained unchanged (*p* > 0.05) (Table 3).

Spearman correlation analysis showed that the difference in the LDL-C level between baseline and 3 months was correlated with the difference in the GI between baseline and 3 months (R = 0.38, *p* = 0.04). However, repeated-measures analysis of variance revealed that changes in periodontal parameters during the 3-month follow-up period did not significantly affect changes in the measured serum parameters.

## 4. Discussion

Our clinical study demonstrated that both non-surgical and surgical dental treatments led to improvements in clinical periodontal parameters (PI, GI, and PPD) and the microflora of dental plaque and sulcus samples, alongside reductions in the serum lipid and CRP levels. At the 3-month follow-up, significant decreases in the serum cholesterol and CRP levels were observed in the hyperlipidemic patient group, and moderate improvement in LDL-C levels was also detected. Although the triglyceride and HDL-C levels improved, these changes were not statistically significant. The novelty of our protocol lies in its inclusion of not only periodontal but also pulpal-origin inflammation in the treatment regimen. This comprehensive approach to dental treatment in hyperlipidemic patients has not been previously described, highlighting the potential systemic benefits of addressing multiple sources of dental inflammation.

At baseline, significant differences were observed in periodontal parameters (PI, GI, PPD) and microbial profiles of dental plaque and sulcus samples between the hyperlipidemic patient group and the healthy control group, consistent with previous findings [10]. The potential associations and bidirectional relationships between hyperlipidemia and dental infections such as periodontitis have been widely investigated [3,17,18,19,20,21,22,23,24,25,26]. The concept of overlapping and interconnected etiopathological mechanisms in dental inflammatory diseases, which may increase the risk of hyperlipidemia and thereby elevate the CVD risk, is well established [30,47]. Previous studies have shown that poor oral hygiene and hyperlipidemia often co-exist [15,17,19,26]. Similarly, our findings align with studies evaluating the periodontal status of hyperlipidemic patients [17,19,20,21,22,29], as supported by Fentoglu et al., with the hyperlipidemic group exhibiting significantly higher periodontal parameters (GI, PI, PPD) than the healthy controls [17]. Despite lipid-lowering therapy in our study, no significant differences were observed in LDL-C or HDL-C levels, a borderline difference was found in total cholesterol levels, and a significant difference was noted in triglyceride levels between the groups at baseline, likely due to the lower efficacy of statin and fibrate therapy in reducing triglyceride levels [48,49,50,51]. Following causative therapy, including scaling and root planing, significant improvements were observed in all indices (PI, GI, and PPD) and in subgingival and supragingival plaque samples, aligning with previous findings [18]. Additionally, a notable shift in the microflora was observed, moving from a dominance of pathogenic bacteria toward a commensal microflora in both plaque and subgingival samples.

Odontogenic infection-related inflammatory signals and effector molecules contributing to dysregulated lipid metabolism have been previously reported [52,53], potentially explaining the observed improvements in serum metabolic parameters following intensive periodontal and endodontic treatments in healthy patients with periodontitis and apical periodontitis [35,54,55]. Earlier studies evaluated the effects of periodontal treatment on serum lipid parameters in both healthy and hyperlipidemic patients. D’Aiuto reported that intensive periodontal therapy was more effective than standard treatments in improving serum lipid parameters [54]. A Turkish research group systematically studied the metabolic and inflammatory effects of lipid-lowering therapy combined with periodontal treatment in hyperlipidemic patients [19,20,56]. In 2010, they demonstrated that 3-month statin therapy significantly improved lipid profiles in hyperlipidemic patients, and subsequent non-surgical periodontal treatment led to further reductions in serum total cholesterol and LDL-C levels. In our study, which incorporated combined non-surgical and surgical dental treatments, significant reductions in serum cholesterol levels and borderline reductions in LDL-C levels were observed, while HDL-C and triglyceride levels remained unchanged at the 3-month follow-up. The differing outcomes between Fentoglu et al. [19] and our study could be attributed to our protocol, which eliminated the short-term effects of lipid-lowering therapy by requiring a strict 6-month regimen of lipid-lowering medication, lifestyle modifications, and dietary adjustments under specialist supervision before initiating dental treatments, as opposed to their shorter, 3-month statin treatment prior to periodontal therapy.

Periodontal and pulpo-periapical infections are associated with systemic microinflammation [57,58]. Poornima et al. reported a significant decrease in CRP levels following RCT in healthy individuals [59]. In our study, serum CRP levels were assessed at baseline and at two follow-up points—1 week and 3 months after completing combined dental treatments. The 1-week follow-up allowed for monitoring rapid changes and immediate effects of the treatments on serum inflammatory parameters, while the 3-month follow-up evaluated their long-term impact [35]. Weight loss, diet, physical activity, and smoking cessation, as well as 6 weeks of statin or fibrate therapy, are known to reduce serum CRP levels [60,61,62,63]. To control these confounding factors, we implemented a strict protocol before initiating combined dental treatments. Our findings demonstrated a significant reduction in serum CRP levels at the 3-month checkpoint, consistent with Tawfig et al., who observed similar decreases in hyperlipidemic patients with chronic periodontitis 3 months after non-surgical periodontal therapy [64]. Conversely, Fentoglu et al. reported no significant changes in CRP levels [20,22], and another study unexpectedly noted CRP elevation following statin therapy [19]. Apical periodontitis has previously been shown to elevate serum CRP levels by inducing low-grade systemic inflammation [65]. Treating apical periodontitis eliminates sources of toxins and bacterial by-products, potentially reducing immune system activation and mitigating systemic inflammation [59].

In this study, correlation analysis revealed that changes in the investigated parameters did not show simple associations but rather influenced each other in a complex manner. The patients included in this study did not have severe periodontitis, which is why characteristic values reflecting periodontal disease activity were analyzed in relation to serum values. Although the potential systemic consequences of inflammatory lesions are not fully understood, their association with systemic parameters and general health has been previously described [65].

### Limitation of the Study

Our study population was relatively small, which can be attributed to the strict inclusion criteria, the requirement for rigorous and continuous specialist control of well-balanced lipid-lowering therapy, the wide range of dental treatments performed, and the need for long-term patient compliance throughout the study period. Further investigations involving a larger patient population, conducted within the framework of a multicenter study, may confirm and expand upon these findings.

## 5. Conclusions

Combined non-surgical and surgical dental treatments targeting local dental inflammations of pulpal and/or periodontal origin are associated with reductions in CRP and cholesterol levels. These findings suggest that such treatments may serve as beneficial adjunct therapy alongside lipid-lowering therapy in patients with hyperlipidemia.

## Figures and Tables

**Figure 1 jcm-14-00241-f001:**
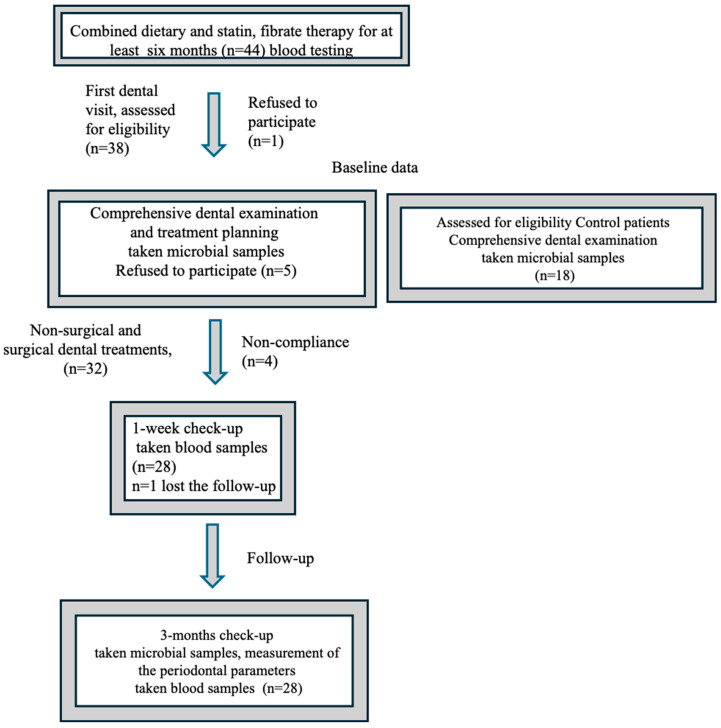
Flow chart of study design.

**Figure 2 jcm-14-00241-f002:**
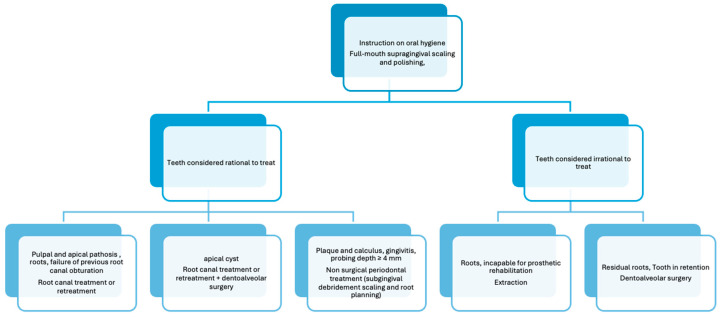
Clinical pathoses detected and treatment modalities performed in the hyperlipidemic patient group during the study period.

**Figure 3 jcm-14-00241-f003:**
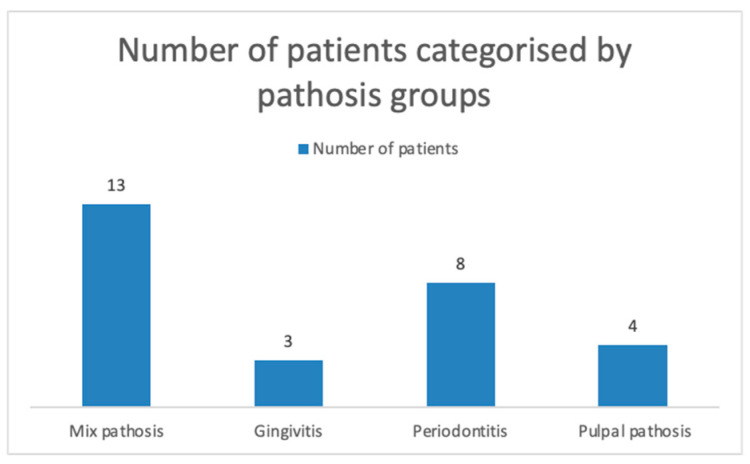
Number of patients categorized by pathosis groups.

**Figure 4 jcm-14-00241-f004:**
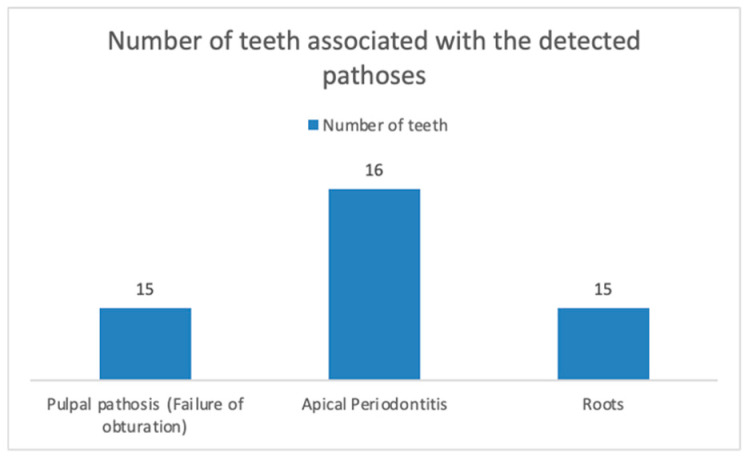
Number of teeth associated with the detected pathoses.

**Figure 5 jcm-14-00241-f005:**
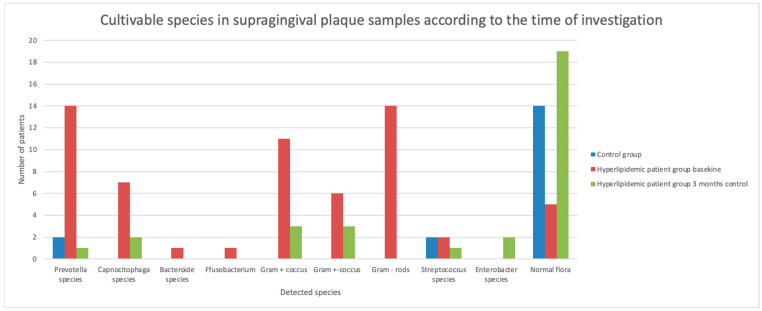
Cultivable species in supragingival plaque samples according to the time of investigation.

**Figure 6 jcm-14-00241-f006:**
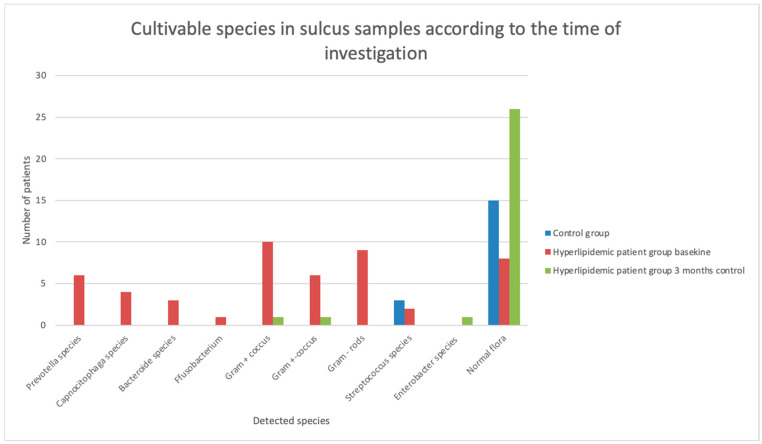
Cultivable species in sulcus samples according to the time of investigation.

**Table 1 jcm-14-00241-t001:** Clinical and periodontal characteristics in the control group and baseline versus 3-month results of hyperlipidemic patients.

			N	Mean [95% Confidence Interval]	SD	Min	Max	Percentile			*p* *
								25th	50th (Median)	75th	(Paired *t*-Test/Independent *t*-Test)
Age		baseline hyperlipidemic patients	28	53.3 [49.0–57.6]	11.60	32.00	71.00	43.50	54.00	63.00	(reference)
		controls for CRP evaluation	28	52.1 [48.1–56.0]	10.69	32.00	71.00	44.50	54.50	58.75	0.677 **
		healthy control	18	51.2 [45.7–56.6]	11.80	32.00	74.00	41.50	50.50	61.25	0.548
Body Mass Index		baseline hyperlipidemic patients	28	30.2 [27.8–32.6]	6.40	18.97	43.75	25.66	29.95	35.18	(reference)
		healthy control	18	26.0 [24.1–28.0]	4.24	20.00	36.14	23.86	25.15	27.64	0.019
Missing teeth		baseline hyperlipidemic patients	28	10.3 [7.89–12.7]	6.56	0.00	23.00	5.00	9.00	13.75	(reference)
		healthy control	18	7.39 [4.31–10.4]	6.66	0.00	25.00	1.25	7.00	9.75	0.148
Periodontal status	PI	baseline hyperlipidemic patients	28	1.02 [0.89–1.16]	0.37	0.42	1.75	0.69	1.04	1.24	(reference)
		3 months after treatment	28	0.73 [0.55–0.90]	0.47	0.00	1.80	0.31	0.70	1.03	<0.001
		healthy control	18	0.34 [0.25–0.44]	0.20	0.13	0.91	0.20	0.31	0.43	<0.001
	GI	baseline hyperlipidemic patients	28	0.71 [0.52–0.91]	0.52	0.10	1.67	0.26	0.59	1.11	(reference)
		3 months after treatment	28	0.55 [0.40–0.69]	0.39	0.02	1.42	0.31	0.42	0.81	0.034
		healthy control	18	0.25 [0.14–0.37]	0.25	0.02	0.90	0.07	0.20	0.31	<0.001
	PPD	baseline hyperlipidemic patients	28	2.02 [1.82–2.22]	0.54	1.05	3.16	1.56	2.00	2.42	(reference)
		3 months after treatment	28	1.73 [1.58–1.88]	0.42	1.10	2.73	1.39	1.71	2.04	0.001
		healthy control	18	1.50 [1.39–1.62]	0.25	1.15	2.08	1.27	1.54	1.62	<0.001

* Differences between patients’ baseline and healthy controls’ periodontal status were analyzed using independent *t*-tests. The treatment effect (baseline vs. 3-month results) for periodontal disease was evaluated using paired *t*-tests. ** Differences in CRP serum levels between patients’ baseline and healthy controls’ for CRP evaluation data.

**Table 2 jcm-14-00241-t002:** Types of subgingival and supragingival flora at baseline and 3-month samples.

Type of Examination		Type of the Examined Flora		*p*
		Normal (%)	Pathogenic (%)	
Supragingival plaque	baseline	15.38	84.62	(reference)
	3 months after treatment	67.86	32.14	<0.001
	healthy control	77.78	22.22	<0.001
Subgingival plaque	baseline	26.92	73.08	(reference)
	3 months after treatment	92.86	7.14	<0.001
	healthy control	83.33	16.67	<0.001

**Table 3 jcm-14-00241-t003:** Distributions of laboratory findings in the studied sample.

Type of Examinations		N	Mean [95% Confidence Interval]	SD	Min	Max	Percentile			*p* *		*p* *	
							25th	50th (Median)	75th	One-Way ANOVA	Friedman Test	Paired *t*-Test/Independent *t*-Test	Wilcoxon Test/Mann-Whitney Test
CRP mg/L	baseline	28	3.30 [2.21–4.39]	2.94	0.63	13.81	1.25	2.54	4.52	0.020	-	(reference)	
	1 week after treatment	27	3.12 [2.26–3.97]	2.26	0.50	9.01	1.23	3.03	4.26			0.928	-
	3 months after treatment	28	2.29 [1.65–2.94]	1.73	0.40	6.38	0.93	1.89	3.34			0.047	-
	healthy control	18	1.69 [1.22–2.16]	2.16	0.50	7.70	0.61	1.15	2.61			0.14	-
	control for CRP evaluation	28	2.09 [1.09–3.09]	1.28	0.16	4.16	0.66	1.43	2.87			0.011	-
Triglyceride mmol/L	baseline	28	2.93 [1.83–4.04]	2.97	0.67	12.55	1.27	1.90	2.85	-	0.884	(reference)	
	1 week after treatment	27	2.46 [1.73–3.19]	1.95	0.84	7.51	1.28	1.60	3.50			-	0.393
	3 months after treatment	28	2.22 [1.59–2.85]	1.70	0.56	6.90	1.25	1.67	2.20			-	0.292
	healthy control	18	1.42 [1.01–1.82]	0.88	0.40	4.20	0.93	1.15	1.60			-	0.009
LDL-C mmol/L	baseline	28	3.71 [3.11–4.31]	1.61	1.67	8.58	2.73	3.30	4.35	0.603	-	(reference)	
	1 week after treatment	24	3.33 [2.81–3.85]	1.30	1.40	6.90	2.42	3.43	3.90			0.327	-
	3 months after treatment	27	3.35 [2.76–3.94]	1.56	0.53	7.60	2.39	3.20	4.00			0.057	-
	healthy control	18	3.20 [2.89–3.51]	0.68	1.80	4.60	2.85	3.20	3.40			0.161	-
Cholesterol mmol/L	baseline	28	6.24 [5.62–6.87]	1.69	4.50	11.36	5.20	5.69	6.90	-	0.054	(reference)	
	1 week after treatment	27	5.77 [5.21–6.32]	1.48	3.57	9.65	4.77	5.64	6.62			-	0.146
	3 months after treatment	28	5.71 [5.15–6.27]	1.51	3.63	9.90	4.77	5.31	6.60			-	0.007
	healthy control	18	5.24 [4.88–5.61]	0.80	3.60	6.80	4.83	5.30	5.58			-	0.051
HDL-C mmol/L	baseline	28	1.46 [1.30–1.62]	0.43	0.80	2.38	1.12	1.36	1.79	0.486	-	(reference)	
	1 week after treatment	27	1.38 [1.25–1.52]	0.36	0.80	2.07	1.10	1.21	1.80			0.017	-
	3 months after treatment	28	1.51 [1.36–1.66]	0.40	1.00	2.26	1.20	1.50	1.96			0.428	-
	healthy control	18	1.58 [1.43–1.74]	0.34	0.90	2.20	1.50	1.60	1.78			0.308	-

* Normality was assessed using the Kolmogorov–Smirnov test.

## Data Availability

The original contributions presented in this study are included in the article. Further inquiries can be directed to the corresponding author.

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
