# Peer review of "Effects of Combined Periodontal, Endodontic, and Dentoalveolar Surgical Treatments on Laboratory Parameters in Patients with Hyperlipidemia—A Clinical Interventional Study"

_jcm, 2025, doi:10.3390/jcm14010241_

Round 1

Reviewer 1 Report

Comments and Suggestions for Authors

Although the article addresses a relevant topic, it lacks originality and does not make significant contributions to the scientific literature. The results are described in an extremely speculative manner, highlighting, for example, that improved oral health led to normalization of sub- and supragingival plaque flora, something already widely known. In addition, it is stated that this improvement contributed to the reduction of C-reactive protein and cholesterol levels, consequently reducing the risk of cardiovascular diseases and suggesting a beneficial effect of oral health indices in the treatment of patients with hyperlipidemia. However, the statistical tests used only assess associations between variables, without demonstrating causality, which compromises the validity of these conclusions.

The title also deserves to be reformulated. It is recommended to remove the expression "Treatment of dental inflammatory lesions" and keep only "Effects on the laboratory parameters in patients with hyperlipidemia: a clinical interventional study", which would make the title more concise and aligned with the focus of the study.

In the introduction, the authors need to clarify the real reason for conducting the study. What specific gap in the literature does this paper aim to fill? The role of oral health in systemic health is already widely recognized, and it is essential to justify how this study adds to existing knowledge.

The text is excessively long and contains many abbreviations, some of which were not previously defined, making it difficult to read.

The tables are confusing; for example, the meaning of the term "reference" needs to be explained clearly.

The methodology is also a critical point. To ensure replicability, the procedures described must be detailed with greater rigor. One example is the section "Assessment of microbial growth": the simple mention that the bacteria were identified based on Gram-staining, colony morphology and biochemical tests is insufficient. The authors need to specify which methods and protocols were used, with sufficient detail to allow the study to be reproduced.

The discussion requires significant reformulation. It should be conducted in light of current scientific knowledge, contrasting the results obtained with the relevant literature. It is important to include possible limitations of the study and suggest future directions for research in this field, which would add value and depth to the work.

Author Response

Reviewer 1

We appreciate a lot the useful comments and suggestions of the peer-reviewers to our manuscript entitled “Treatment of dental inflammatory lesions: effects on the laboratory parameters in patients with hyperlipidemia: a clinical interventional study (jcm-3352978)” by R. Martos et al. Below we would like to reply to the reviewers’ comments in a point-by-point fashion. Accordingly, we have introduced changes in the revised manuscript that are highlighted in yellow (in the revised manuscript). Some alterations in the revised manuscript represent a combination of the responses to more than one reviewer out of the three reviewers.

Comments and Suggestions for Authors

Although the article addresses a relevant topic, it lacks originality and does not make significant contributions to the scientific literature. The results are described in an extremely speculative manner, highlighting, for example, that improved oral health led to normalization of sub- and supragingival plaque flora, something already widely known. In addition, it is stated that this improvement contributed to the reduction of C-reactive protein and cholesterol levels, consequently reducing the risk of cardiovascular diseases and suggesting a beneficial effect of oral health indices in the treatment of patients with hyperlipidemia. However, the statistical tests used only assess associations between variables, without demonstrating causality, which compromises the validity of these conclusions.

The title also deserves to be reformulated. It is recommended to remove the expression "Treatment of dental inflammatory lesions" and keep only "Effects on the laboratory parameters in patients with hyperlipidemia: a clinical interventional study", which would make the title more concise and aligned with the focus of the study.

Thank you for the suggestion. We modified the title according to the recommended reformulation.

The new title is:

 Effects on the laboratory parameters in patients with hyperlipidemia: a clinical interventional study

In the introduction, the authors need to clarify the real reason for conducting the study. What specific gap in the literature does this paper aim to fill? The role of oral health in systemic health is already widely recognized, and it is essential to justify how this study adds to existing knowledge.

Thank you for your comments. We added a new Introduction section to summarise the scientific background of the topic and explain more understandable way novelty of our study.

Introduction

The possible interplay between chronic local inflammation and systemic diseases has been highlighted by review articles summarising longitudinal observational studies, cross-sectional studies, and case-control studies [1–4] Cardiovascular disease (CVD) is a leading cause of mortality worldwide each year [5], and it has been associated with dental infections [6,7]. Increasing attention has been drawn to the observation that a higher frequency of dental infections, such as endodontic infections, gingivitis, and periodontitis; a higher number of missing teeth; and poorer oral health have been identified in patients with chronic heart disease, acute myocardial infarction, or CVD across various populations compared with healthy control subjects [1,8–15]. Furthermore, it is worth noting that among health-conscious patients and those of higher socioeconomic status, root canal treatment (RCT) is often preferred over extraction. Therefore, in addition to missing teeth, the number of RCTs should also be considered when assessing patients’ risk of oral inflammation-related CVD [7,11].

Hyperlipidaemia is a well-known risk factor for future CVD events [16], but the periodontal condition can also influence the risk of CVD [3,17–26]. The systemic effects of plaque-induced periodontal disease and its treatment on the serum C-reactive protein (CRP) level, as well as the potential association between severe periodontal disease and hyperlipidaemia, have been previously described [1,2,15,17–19,24,26–30]. Several studies have revealed that poor oral hygiene and hyperlipidaemia often co-exist, ultimately leading to an increased risk of developing CVD [15,17,19,26] Consequently, multiple studies conducted over the last decade have evaluated the periodontal condition of hyperlipidaemic patients [17,19–22,29] and the efficacy of various periodontal therapies in hyperlipidaemic patients with periodontitis [19–22] .

Examination of the possible interplay between apical periodontitis and systemic diseases has evidenced a moderate risk for CVD, although this relationship is influenced by several confounding factors. Predicting the effects of endodontic treatment in isolation remains challenging [31,32]. The local and systemic immunological effects of chronic apical periodontitis [33,34] and the impact of combined RCT and surgical treatment on serum CRP levels in populations with good general health have been previously reported [35,36]. The systemic effects may result from coexisting pulpal and chronic apical inflammation, triggered by anaerobic bacteria. This process, similar to plaque-induced periodontal disease, is associated with cytokine production and tissue destruction [37–39]. Furthermore, low-density lipoprotein (LDL) may interact with Toll-like receptor 4, similarly to how lipopolysaccharide (LPS) from the bacterial walls of periodontopathogenic bacteria functions, while Toll-like receptor 2 can also interact with LPS from Gram-positive bacteria and bind LDL [40]. This could represent an important pathway linking periodontal disease and hyperlipidaemia.

The systemic effects of concurrently treating pulpal and periapical inflammation in hyperlipidaemic patients have not been previously evaluated. Therefore, our interventional clinical study aimed to assess the impact of combined periodontal, endodontic, and dentoalveolar surgical treatments on serum CRP levels and serum lipid components and lipoproteins in patients with hyperlipidaemia. To the best of our knowledge, evidence on the effects of these treatment modalities on laboratory parameters in hyperlipidaemic patient groups is lacking. We hypothesised that in addition to the beneficial effects of treating periodontal disease, eliminating microbial sources associated with pulpal inflammation could provide adjunctive therapeutic support for hyperlipidaemic patients.

The text is excessively long and contains many abbreviations, some of which were not previously defined, making it difficult to read.

Thank you for your comments. We tried to describe the whole study more concisely, unfortunately, regarding the complex nature of the recruitment, examinations, and treatment protocols we tried to explain all the sufficient information. We added all the definitions of the abbreviations.

The tables are confusing; for example, the meaning of the term "reference" needs to be explained clearly.

Thank you for your comment! We modified the tables according tot he required explanation.

 Control group

Age- and sex-matched healthy controls willing to participate in this study were selected from patients attending check-ups at the Department of Restorative Dentistry, Faculty of Dentistry, MHSC UD. Controls were selected to exclude individuals with hyperlipidaemia or hypertension, and smokers were also excluded. Body mass index was calculated by a physician.

Age- and sex-matched healthy controls (n = 28; mean age, 52.11 ± 10.69 years; 10 men and 18 women) for CRP evaluation were selected from the control pool of the TECH-09-A1-2009-0113 (mAB-CHIC) project of the Department of Laboratory Medicine, MHSC UD.

The methodology is also a critical point. To ensure replicability, the procedures described must be detailed with greater rigor. One example is the section "Assessment of microbial growth": the simple mention that the bacteria were identified based on Gram-staining, colony morphology and biochemical tests is insufficient. The authors need to specify which methods and protocols were used, with sufficient detail to allow the study to be reproduced.

Thank you for your suggestions, we added the more detailed material and methods subsection. Modified text part highlighted with yellow in the text.

Microbiological examination

Microbial examinations were conducted using subgingival and supragingival plaque samples collected before and 3 months after complex dental treatments. Bacterial samples were taken from the site with the deepest probing depth, as identified by the same dentist during the initial examination, and the identical site was re-examined 3 months after the combined dental treatments.

Under cotton roll isolation, sterile paper points were inserted with tweezers into the apical extent of the deepest sites. The absorbent points were then removed with tweezers, placed into a test tube, and immediately transferred to a sterile transport medium (Stuart transport medium, Remel, Lenexa, US) using the swab. Supragingival plaque culture was collected with a special sterile transport swab (Stuart transport medium). The test tubes were transported to the Bacteriology Laboratory of the Department of Medical Microbiology, MHSC UD, where the samples were processed according to standard laboratory methods. The presence of bacterial species was detected non-quantitatively, indicating whether each species was present or absent. The distribution of pathogenic versus non-pathogenic subgingival and supragingival plaque flora was expressed as a proportional value of all detected species.

Laboratory culturing method

Under strict aseptic conditions, a calibrated (10 µL) platinum loop was used to streak the homogenously dispersed material from the test tubes onto two Petri plates containing Columbia agar supplemented with 5% sheep blood, hemin (5 mg/L), and 0.4 µL/mL vitamin K1 (bioMérieux, Lyon, France). One Petri plate was incubated under aerobic conditions, while the other was incubated anaerobically for 5 days in a Ruskinn Concept 400 anaerobic chamber (Labtech, Rotherham, UK).

Assessment of microbial growth

Microbial growth in the Petri plates was assessed after 5 days. Bacteria were identified based on Gram staining, colony morphology, biochemical tests (e.g., catalase, coagulase, oxidase), and the VITEK 2 automated identification system (bioMérieux, Lyon, France), following the manufacturer’s instructions. The VITEK 2 system operates with specific cards for the identification of Gram-negative,Gram-positive, and anaerobic bacteria. Bacterial or fungal colonies were suspended in a 0.45% sodium chloride medium, withsuspension turbidity adjusted to 0.5 McFarland for Gram-negative bacteria, 1.0 McFarland for Gram-positive bacteria, and 3.0McFarland for anaerobic bacteria. The inoculums were introduced into the corresponding VITEK 2 cards and incubated for 18 to 20 hours. The identification time varied depending on bacterial metabolism and the card type. 

The discussion requires significant reformulation. It should be conducted in light of current scientific knowledge, contrasting the results obtained with the relevant literature. It is important to include possible limitations of the study and suggest future directions for research in this field, which would add value and depth to the work.

Thank you for the suggestions, we added a new discussion section and also the limitation of the study section.

Discussion

Our clinical study demonstrated that both non-surgical and surgical dental treatments led to improvements in clinical periodontal parameters (PI, GI, and PPD) and the microflora of dental plaque and sulcus samples, alongside reductions in the serum lipid and CRP levels. At the 3-month follow-up, significant decreases in the serum cholesterol and CRP levels were observed in the hyperlipidaemic patient group, and moderate improvement in LDL-C levels was also detected. Although the triglyceride and HDL-C levels improved, these changes were not statistically significant. The novelty of our protocol lies in its inclusion of not only periodontal but also pulpal-origin inflammation in the treatment regimen. This comprehensive approach to dental treatment in hyperlipidaemic patients has not been previously described, highlighting the potential systemic benefits of addressing multiple sources of dental inflammation.

At baseline, significant differences were observed in periodontal parameters (PI, GI, PPD) and microbial profiles of dental plaque and sulcus samples between the hyperlipidaemic patient group and the healthy control group, consistent with previous findings [10]. The potential associations and bidirectional relationships between hyperlipidaemia and dental infections such as periodontitis have been widely investigated [3,17–26]. The concept of overlapping and interconnected aetiopathological mechanisms in dental inflammatory diseases, which may increase the risk of hyperlipidaemia and thereby elevate the CVD risk, is well established [30,47]. Previous studies have shown that poor oral hygiene and hyperlipidaemia often co-exist [15,17,19,26]. Similarly, our findings align with studies evaluating the periodontal status of hyperlipidaemic patients [17,19–22,29], as supported by Fentoglu et al., with the hyperlipidaemic group exhibiting significantly higher periodontal parameters (GI, PI, PPD) than the healthy controls [17]. Despite lipid-lowering therapy in our study, no significant differences were observed in LDL-C or HDL-C levels, a borderline difference was found in total cholesterol levels, and a significant difference was noted in triglyceride levels between the groups at baseline, likely due to the lower efficacy of statin and fibrate therapy in reducing triglyceride levels [48–51]. Following causative therapy, including scaling and root planing, significant improvements were observed in all indices (PI, GI, and PPD) and in subgingival and supragingival plaque samples, aligning with previous findings [18]. Additionally, a notable shift in the microflora was observed, moving from a dominance of pathogenic bacteria toward a commensal microflora in both plaque and subgingival samples.

Odontogenic infection-related inflammatory signals and effector molecules contributing to dysregulated lipid metabolism have been previously reported [52,53], potentially explaining the observed improvements in serum metabolic parameters following intensive periodontal and endodontic treatments in healthy patients with periodontitis and apical periodontitis [54–56]. Earlier studies evaluated the effects of periodontal treatment on serum lipid parameters in both healthy and hyperlipidaemic patients. D’Aiuto reported that intensive periodontal therapy was more effective than standard treatments in improving serum lipid parameters [54]. A Turkish research group systematically studied the metabolic and inflammatory effects of lipid-lowering therapy combined with periodontal treatment in hyperlipidaemic patients [19,20,57]. In 2010, they demonstrated that a 3-month statin therapy significantly improved lipid profiles in hyperlipidaemic patients, and subsequent non-surgical periodontal treatment led to further reductions in serum total cholesterol and LDL-C levels. In our study, which incorporated combined non-surgical and surgical dental treatments, significant reductions in serum cholesterol levels and borderline reductions in LDL-C levels were observed, while HDL-C and triglyceride levels remained unchanged at the 3-month follow-up. The differing outcomes between Fentoglu et al. [19] and our study could be attributed to our protocol, which eliminated the short-term effects of lipid-lowering therapy by requiring a strict 6-month regimen of lipid-lowering medication, lifestyle modifications, and dietary adjustments under specialist supervision before initiating dental treatments, as opposed to their shorter, 3-month statin treatment prior to periodontal therapy.

Periodontal and pulpo-periapical infections are associated with systemic microinflammation [58,59]. Poornima et al. reported a significant decrease in CRP levels following RCT in healthy individuals [60]. In our study, serum CRP levels were assessed at baseline and at two follow-up points—1 week and 3 months after completing combined dental treatments. The 1-week follow-up allowed for monitoring rapid changes and immediate effects of the treatments on serum inflammatory parameters, while the 3-month follow-up evaluated their long-term impact [35]. Weight loss, diet, physical activity, and smoking cessation, as well as 6 weeks of statin or fibrate therapy, are known to reduce serum CRP levels [61–64]. To control these confounding factors, we implemented a strict protocol before initiating combined dental treatments. Our findings demonstrated a significant reduction in serum CRP levels at the 3-month checkpoint, consistent with Tawfig et al., who observed similar decreases in hyperlipidaemic patients with chronic periodontitis 3 months after non-surgical periodontal therapy [65]. Conversely, Fentoglu et al. reported no significant changes in CRP levels [20,22], and another study unexpectedly noted CRP elevation following statin therapy [19]. Apical periodontitis has previously been shown to elevate serum CRP levels by inducing low-grade systemic inflammation [66]. Treating apical periodontitis eliminates sources of toxins and bacterial by-products, potentially reducing immune system activation and mitigating systemic inflammation [60].

.

In this study, correlation analysis revealed that changes in the investigated parameters did not show simple associations but rather influenced each other in a complex manner. The patients included in this study did not have severe periodontitis, which is why characteristic values reflecting periodontal disease activity were analysed in relation to serum values. Although the potential systemic consequences of inflammatory lesions are not fully understood, their association with systemic parameters and general health has been previously described [66].

Limitation of the study

Our study population was relatively small, which can be attributed to the strict inclusion criteria, the requirement for rigorous and continuous specialist control of well-balanced lipid-lowering therapy, the wide range of dental treatments performed, and the need for long-term patient compliance throughout the study period. Further investigations involving a larger patient population, conducted within the framework of a multicentre study, may confirm and expand upon these findings.

Conclusion

Combined non-surgical and surgical dental treatments targeting local dental inflammations of pulpal and/or periodontal origin are associated with reductions in CRP and cholesterol levels. These findings suggest that such treatments may serve as a beneficial adjunct therapy alongside lipid-lowering therapy in patients with hyperlipidaemia.

Reviewer 2 Report

Comments and Suggestions for Authors

An ABSTRACT has more than 250 words

·        A detailed description of the statistical test certainly does not belong in the abstract

INTRODUCTION:

·        Correct typos

·        Line 62: Aterosclerosis is not an ethological factor, but a pathophysiological process

·        Line 63: oral infections are not an appropriate description for most oral diseases; they are rather a consequence of dysbiosis due to ecological disturbances in oral biofilms

·        Line 82-94: the peculiar odour

·        Line 85: what kind of lipid-lowering therapy?

The AIM is confused

HYPOTHESIS is missing.

MATERIAL AND METHODS

·        Clinical examination:

·        what type of periodontal probe was used (brand, type)

·        the EFP-based criteria for the diagnosis of periodontitis must be used

·        GI is not related to gingival bleeding, please correct

·        It is not clear how the reproducibility of PI and GI was assessed

·        It appears that the bacteria were evaluated by cultivation methods. The transport media, cultivation conditions and especially the identification methods must be described in sufficient detail.

·        Treatment: Describe briefly but in sufficient detail each type of treatment performed

·        Dental treatments: Use standard terms to describe the procedure: e.g. line 212: scaling and root planing

·        Sample size calculation: I understand that the CRP difference of 1 mmol/l was used to calculate the sample size. What measure of variability was used? The sample size calculation is not clear.

·        The study design is completely strange. This study cannot be defined as an intervention study because no two treatment protocols were compared in the test and control groups to calculate the probability of CRP lowering between the groups depending on the therapy administered. On the other hand, it is also not an observational study with two groups, as the treatment was administered in both groups. Therefore, the authors must make clear what the basic study design of the reported research project is. Only then is a clear presentation of the hypotheses and results possible. Therefore, a complete restatement of the results is required.

RESULTS

·        As this is an intervention study, a CONSORT flowchart and CONSORT table must be included.

·        The experimental and control groups must be matched in size.

·        The results should be abbreviated and presented in accordance with the CONSORT recommendations. Now it is very difficult to follow the results as the text is full of errors and written in uncorrected English.

·        As written, 15/28 patients in the test group were without periodontitis. The frequency of all dental pathologies should be reported, including the frequency and details of patients with mixed pathology. These data should be listed in the table.

·        The frequency of all bacteria should be reported in a table. As this is not generally accepted, the distinction between pathogenic and normal species needs to be removed.

·        The odds for lowering CRP as the primary outcome should be calculated in relation to the different terapeutic measures.

DISCUSSION

·        The first paragraph should summarise the most important results

·        The authors must place their results in the context of other, similar research data

Comments on the Quality of English Language

I suggest that you also correct the article linguistically, as it is difficult to read.

Author Response

Reviewer 2

 We appreciate a lot the useful comments and suggestions to our manuscript entitled “Treatment of dental inflammatory lesions: effects on the laboratory parameters in patients with hyperlipidemia: a clinical interventional study (jcm-3352978)” by R. Martos et al. Below we would like to reply to the comments in a point-by-point fashion. Accordingly, we have introduced changes in the revised manuscript that are highlighted in yellow (in the revised manuscript). Some alterations in the revised manuscript represent a combination of the responses to more than one reviewer out of the three reviewers.

An ABSTRACT has more than 250 words

  • A detailed description of the statistical test certainly does not belong in the abstract

Thank you for your comment, we removed from the abstract thedescription of the statistical test to reduce the word count up to 250 words.

Background: Patients with hyperlipidaemia are of interest because of the possible interplay between chronic local dental infections and hyperlipidaemia. This interventional clinical study aimed to evaluate the oral health status of hyperlipidaemic patients receiving lipid-lowering therapy for at least 6 months and the effects of non-surgical and surgical dental treatments on serum C-reactive protein (CRP) levels and lipid markers.

Methods: Twenty-eight patients with controlled hyperlipidaemia and 18 healthy controls were enrolled in the study. All participants underwent dental examinations (clinical evaluation, X-ray imaging, and microbial analysis of subgingival and supragingival plaque samples) at baseline. Hyperlipidaemic patients received periodontal, endodontic, and dentoalveolar surgical treatments. Serum CRP and lipid parameters were assessed at baseline, 1 week, and 3 months, while subgingival and supragingival plaque samples were analysed at baseline and 3 months after completing dental treatments.

Results: At the 3-month follow-up, clinical periodontal characteristics, including the plaque index, gingival index, and periodontal probing depth, improved significantly (p < 0.05). A significant shift in microflora was observed in both subgingival and supragingival plaque samples (p < 0.05), alongside improvements in periodontal values and a significant reduction in serum CRP levels (p < 0.05). Serum cholesterol levels decreased significantly, while moderate improvements in serum triglycerides, low-density lipoprotein, and high-density lipoprotein levels were observed but were not statistically significant (p > 0.05).

Conclusions: Treating local dental inflammation is associated with a significant decrease in CRP and cholesterol levels and may serve as a beneficial adjunct therapy alongside lipid-lowering therapy in patients with hyperlipidaemia.

INTRODUCTION:

  • Correct typos
  • Line 62: Aterosclerosis is not an ethological factor, but a pathophysiological process
  • Line 63: oral infections are not an appropriate description for most oral diseases; they are rather a consequence of dysbiosis due to ecological disturbances in oral biofilms
  • Line 82-94: the peculiar odour

Thank you for your comments, we added new introduction section.

Introduction

The possible interplay between chronic local inflammation and systemic diseases has been highlighted by review articles summarising longitudinal observational studies, cross-sectional studies, and case-control studies [1–4] Cardiovascular disease (CVD) is a leading cause of mortality worldwide each year [5], and it has been associated with dental infections [6,7]. Increasing attention has been drawn to the observation that a higher frequency of dental infections, such as endodontic infections, gingivitis, and periodontitis; a higher number of missing teeth; and poorer oral health have been identified in patients with chronic heart disease, acute myocardial infarction, or CVD across various populations compared with healthy control subjects [1,8–15]. Furthermore, it is worth noting that among health-conscious patients and those of higher socioeconomic status, root canal treatment (RCT) is often preferred over extraction. Therefore, in addition to missing teeth, the number of RCTs should also be considered when assessing patients’ risk of oral inflammation-related CVD [7,11].

Hyperlipidaemia is a well-known risk factor for future CVD events [16], but the periodontal condition can also influence the risk of CVD [3,17–26]. The systemic effects of plaque-induced periodontal disease and its treatment on the serum C-reactive protein (CRP) level, as well as the potential association between severe periodontal disease and hyperlipidaemia, have been previously described [1,2,15,17–19,24,26–30]. Several studies have revealed that poor oral hygiene and hyperlipidaemia often co-exist, ultimately leading to an increased risk of developing CVD [15,17,19,26] Consequently, multiple studies conducted over the last decade have evaluated the periodontal condition of hyperlipidaemic patients [17,19–22,29] and the efficacy of various periodontal therapies in hyperlipidaemic patients with periodontitis [19–22] .

Examination of the possible interplay between apical periodontitis and systemic diseases has evidenced a moderate risk for CVD, although this relationship is influenced by several confounding factors. Predicting the effects of endodontic treatment in isolation remains challenging [31,32]. The local and systemic immunological effects of chronic apical periodontitis [33,34] and the impact of combined RCT and surgical treatment on serum CRP levels in populations with good general health have been previously reported [35,36]. The systemic effects may result from coexisting pulpal and chronic apical inflammation, triggered by anaerobic bacteria. This process, similar to plaque-induced periodontal disease, is associated with cytokine production and tissue destruction [37–39]. Furthermore, low-density lipoprotein (LDL) may interact with Toll-like receptor 4, similarly to how lipopolysaccharide (LPS) from the bacterial walls of periodontopathogenic bacteria functions, while Toll-like receptor 2 can also interact with LPS from Gram-positive bacteria and bind LDL [40]. This could represent an important pathway linking periodontal disease and hyperlipidaemia.

The systemic effects of concurrently treating pulpal and periapical inflammation in hyperlipidaemic patients have not been previously evaluated. Therefore, our interventional clinical study aimed to assess the impact of combined periodontal, endodontic, and dentoalveolar surgical treatments on serum CRP levels and serum lipid components and lipoproteins in patients with hyperlipidaemia. To the best of our knowledge, evidence on the effects of these treatment modalities on laboratory parameters in hyperlipidaemic patient groups is lacking. We hypothesised that in addition to the beneficial effects of treating periodontal disease, eliminating microbial sources associated with pulpal inflammation could provide adjunctive therapeutic support for hyperlipidaemic patients.

  • Line 85: what kind of lipid-lowering therapy?

Thank you for your question. We added a detailed description into the Materials and method section, Participant subsection.

The administered lipid lowering therapy was the following :Simvastatin (20-40 mg/d), atorvastatin (20-40 mg/d), fluvastatin (80 mg/d) or rosuvastatin (10-40 mg/d) monotherapy or in combination with ezetimibe (10 mg/d) and/or fenofibrate (160 mg/d) was administered according to the current therapeutic guidelines.

The main goal of the delayed enrollment is to exclude the possible confounding effect of lipid-lowering therapy. Therefore, the types and doses of the lipid-lowering agents were kept constant throughout the study. To eliminate the short-term effects of lipid-lowering therapy, we enrolled patients with hyperlipidemia after the adjustment of the combined dietary and statin/fibrate therapy for at least six months. Therefore, the confounding effect of these agents on lipid and inflammatory parameters could be excluded. 

The following section was inserted into the text

Type IIa, also known as familial hypercholesterolaemia, occurs due to a gene alteration and causes high levels of LDL cholesterol (LDL-C) (total cholesterol > 5.2 mmol/L, LDL-C > 3.0 mmol/L). Type IIb is a subvariation involving a gene alteration that causes both high cholesterol and high triglycerides (mixed hyperlipidaemia) (total cholesterol > 5.2 mmol/L, LDL-C > 3.0 mmol/L, triglycerides > 1.7 mmol/L) [41]. Lipid-lowering therapy included simvastatin (20–40 mg/day), atorvastatin (20–40 mg/day), fluvastatin (80 mg/day), or rosuvastatin (10–40 mg/day) as monotherapy or in combination with ezetimibe (10 mg/day) and/or fenofibrate (160 mg/day), administered according to current therapeutic guidelines.

The AIM is confused

Thank you for your comment, we add an aim subsection into the introduction section.

HYPOTHESIS is missing.

Thank you for your comment, we add an hypothesis subsection into the introduction section.

Therefore, our interventional clinical study aimed to assess the impact of combined periodontal, endodontic, and dentoalveolar surgical treatments on serum CRP levels and serum lipid components and lipoproteins in patients with hyperlipidaemia. To the best of our knowledge, evidence on the effects of these treatment modalities on laboratory parameters in hyperlipidaemic patient groups is lacking. We hypothesised that in addition to the beneficial effects of treating periodontal disease, eliminating microbial sources associated with pulpal inflammation could provide adjunctive therapeutic support for hyperlipidaemic patients.

MATERIAL AND METHODS

  • Clinical examination:
  • what type of periodontal probe was used (brand, type)

UNC probe 15 (HU-Friedy, Milano, Italy)

  • the EFP-based criteria for the diagnosis of periodontitis must be used

The diagnosis of periodontitis is well-defined in the literature. Based on the classification : (https://www.efp.org/fileadmin/uploads/efp/Documents/Campaigns/New_Classification/Guidance_Notes/report-02.pdf)

A patient is a periodontitis case when:

  • Interdental CAL is detectable at ≥2 non-adjacent teeth, or
  • Buccal/oral CAL of ≥3mm with pocketing of >3mm is detectable at ≥2 teeth, and
  • The observed CAL cannot be ascribed to non-periodontal causes such as:
  1. Gingival recession of traumatic origin;
  2. Dental caries extending in the cervical area of the tooth;
  3. The presence of CAL on the distal aspect of a second molar and associated with malposition or extraction of a third molar;
  4. An endodontic lesion draining through the marginal periodontium;
  5. The occurrence of a vertical root fracture. Measuring CAL Given the measurement error of CAL with a standard periodontal probe, a degree of misclassification of the initial stage of periodontitis is inevitable.

However based on the material and methods section we disclose participant with severe generalized periodontitis. Similarly to a recetly published article, we defined periodontitis as having PPD> 4 mm. https://link.springer.com/content/pdf/10.1186/s12903-023-03668-7.pdf

  • GI is not related to gingival bleeding, please correct

Originally it was developed for the scoring of the severity of gingivitis. GI is only related to gingival bleeding if it takes 2 or 3.

GI

colour change

bleeding on probing

0

-

-

1

+

-

2

+/-

+

3

+/-

+

  • It is not clear how the reproducibility of PI and GI was assessed

Thank you for your comment. Firstly, we ensured the calibration of the examiner:

A single examiner (R.M.) conducted all assessments throughout the study. The calibration process involved repeated clinical periodontal examinations of five patients. Each patient was assessed twice during one visit, with a 1-hour interval between assessments. The mean differences in the scores of examined sites were measured to an accuracy of ±1 mm or ±1 score. Reproducibility was considered acceptable with at least 85% accuracy. No significant differences were observed between the periodontal values (PI, GI, and PPD) from the two assessments.

  • It appears that the bacteria were evaluated by cultivation methods. The transport media, cultivation conditions and especially the identification methods must be described in sufficient detail.

Thank you for your comment. We added new details to the Microbiological examination, Laboratory culturing method, Assessment of microbial growth subsections

Microbiological examination

Microbial examinations were conducted using subgingival and supragingival plaque samples collected before and 3 months after complex dental treatments. Bacterial samples were taken from the site with the deepest probing depth, as identified by the same dentist during the initial examination, and the identical site was re-examined 3 months after the combined dental treatments.

Under cotton roll isolation, sterile paper points were inserted with tweezers into the apical extent of the deepest sites. The absorbent points were then removed with tweezers, placed into a test tube, and immediately transferred to a sterile transport medium (Stuart transport medium, Remel, Lenexa, US) using the swab. Supragingival plaque culture was collected with a special sterile transport swab (Stuart transport medium). The test tubes were transported to the Bacteriology Laboratory of the Department of Medical Microbiology, MHSC UD, where the samples were processed according to standard laboratory methods. The presence of bacterial species was detected non-quantitatively, indicating whether each species was present or absent. The distribution of pathogenic versus non-pathogenic subgingival and supragingival plaque flora was expressed as a proportional value of all detected species.

Laboratory culturing method

Under strict aseptic conditions, a calibrated (10 µL) platinum loop was used to streak the homogenously dispersed material from the test tubes onto two Petri plates containing Columbia agar supplemented with 5% sheep blood, hemin (5 mg/L), and 0.4 µL/mL vitamin K1 (bioMérieux, Lyon, France). One Petri plate was incubated under aerobic conditions, while the other was incubated anaerobically for 5 days in a Ruskinn Concept 400 anaerobic chamber (Labtech, Rotherham, UK).

Assessment of microbial growth

Microbial growth in the Petri plates was assessed after 5 days. Bacteria were identified based on Gram staining, colony morphology, biochemical tests (e.g., catalase, coagulase, oxidase), and the VITEK 2 automated identification system (bioMérieux, Lyon, France), following the manufacturer’s instructions. The VITEK 2 system operates with specific cards for the identification of Gram-negative,Gram-positive, and anaerobic bacteria. Bacterial or fungal colonies were suspended in a 0.45% sodium chloride medium, withsuspension turbidity adjusted to 0.5 McFarland for Gram-negative bacteria, 1.0 McFarland for Gram-positive bacteria, and 3.0McFarland for anaerobic bacteria. The inoculums were introduced into the corresponding VITEK 2 cards and incubated for 18 to 20 hours. The identification time varied depending on bacterial metabolism and the card type. 

  • Treatment: Describe briefly but in sufficient detail each type of treatment performed.

 Dental treatments: Use standard terms to describe the procedure: e.g. line 212: scaling and root planing

Hyperlipidaemic patients received oral hygiene instructions, including motivation and guidance (clarifying periodontal issues and targeting plaque reduction) for tooth brushing and interdental cleaning. Scaling and root planing were performed, involving the removal of major deposits using ultrasound scaling, subgingival surface planing with hand scalers, and polishing. Caries control was addressed with fluoride-containing tooth gels. No antibiotic treatment was administered. All dental treatments were performed by the same dentist [46].

Indeed, surgical periodontal therapy, filling therapy, prosthetic rehabilitation were not incorporated in the dental treatment modalities applied in course of this study. Unfortunately to discribe all treatment protocol may excede the limit of the article. As a reference, for surgical and non-surgical endodontics, surgical therapies we can use the following sources: Ingle's Endodontics Ilan Rotstein, DDS, John I. Ingle, DDS PMPH US.; Contemporary Oral and Maxillofacial Surgery - E-Book James R. Hupp, Myron R. Tucker, Edward Ellis · 2013; file:///Users/renatadrmartos/Downloads/surgical_endodontics_2012.pdf;https://onlinelibrary.wiley.com/doi/epdf/10.1111/aej.12848

.

  • Sample size calculation: I understand that the CRP difference of 1 mmol/l was used to calculate the sample size. What measure of variability was used? The sample size calculation is not clear.

Thank you for your comments. Our correction are highlighted with yellow in the text.

The required sample size was calculated based on the standard deviation of the CRP levels (1.3 mmol/L) among patients without malignant or inflammatory diseases, as estimated by the Department of Laboratory Medicine, with a 1-mmol/L CRP concentration defined as the minimum detectable change. Age- and sex-matched healthy controls (n = 28; mean age, 52.11 ± 10.69 years; 10 men and 18 women) for CRP evaluation were selected from the control pool of the TECH-09-A1-2009-0113 (mAB-CHIC) project of the Department of Laboratory Medicine, MHSC UD. Assuming equal group sizes for controls and patients, a 5% type I error, and 80% statistical power, the required minimum sample size was 27 patients in the diseased group. The CRP control group data was used exclusively for sample size determination.

.

  • The study design is completely strange. This study cannot be defined as an intervention study because no two treatment protocols were compared in the test and control groups to calculate the probability of CRP lowering between the groups depending on the therapy administered. On the other hand, it is also not an observational study with two groups, as the treatment was administered in both groups. Therefore, the authors must make clear what the basic study design of the reported research project is. Only then is a clear presentation of the hypotheses and results possible. Therefore, a complete restatement of the results is required.

Thank you for your comments and we apologies about the misunderstandable description.

In our study we enrolled 28 hyperlipidemic patients as a patient group and 18 age and gender balanced subjects as a control group. You are completely right that we did not compare two treatment protocol. We performed a comprehensive dental clinical examination in both groups to compare the baseline cliinical parameters, which could be a case control study design. However we perform non-surgical and surgical dental interventions exclusively in the hyperlipidemic patients group to eliminate the local microbial sources originated either pulpal or periodontal inflammation. That is why we used the interventional terminology. We compared the baseline data with the 1 week and 3 month follow-up results that is why we can use follow-up terminology.We are sorry that the interpretation of the study design was strange, therefore we add new Figures (Figure 1, Figure 2) and sentences into the material and methods section.

RESULTS

  • As this is an intervention study, a CONSORT flowchart and CONSORT table must be included.

Our study was a non-randomized follow-up clinical study not involving study drugs and to stress our adherence to the items of CONSORT statement, we have added several new pieces of text to the revised manuscript.

Introduction: Scientific background and explanation of racionale

Specific objectives and hypothesis are completed in “Introduction”; show changed text highlighted in yellow.

Scientific background has been explained more in details. “Introduction (See revised text where changes have been highlighted in yellow.)

Introduction

The possible interplay between chronic local inflammation and systemic diseases has been highlighted by review articles summarising longitudinal observational studies, cross-sectional studies, and case-control studies [1–4] Cardiovascular disease (CVD) is a leading cause of mortality worldwide each year [5], and it has been associated with dental infections [6,7]. Increasing attention has been drawn to the observation that a higher frequency of dental infections, such as endodontic infections, gingivitis, and periodontitis; a higher number of missing teeth; and poorer oral health have been identified in patients with chronic heart disease, acute myocardial infarction, or CVD across various populations compared with healthy control subjects [1,8–15]. Furthermore, it is worth noting that among health-conscious patients and those of higher socioeconomic status, root canal treatment (RCT) is often preferred over extraction. Therefore, in addition to missing teeth, the number of RCTs should also be considered when assessing patients’ risk of oral inflammation-related CVD [7,11].

Hyperlipidaemia is a well-known risk factor for future CVD events [16], but the periodontal condition can also influence the risk of CVD [3,17–26]. The systemic effects of plaque-induced periodontal disease and its treatment on the serum C-reactive protein (CRP) level, as well as the potential association between severe periodontal disease and hyperlipidaemia, have been previously described [1,2,15,17–19,24,26–30]. Several studies have revealed that poor oral hygiene and hyperlipidaemia often co-exist, ultimately leading to an increased risk of developing CVD [15,17,19,26] Consequently, multiple studies conducted over the last decade have evaluated the periodontal condition of hyperlipidaemic patients [17,19–22,29] and the efficacy of various periodontal therapies in hyperlipidaemic patients with periodontitis [19–22] .

Examination of the possible interplay between apical periodontitis and systemic diseases has evidenced a moderate risk for CVD, although this relationship is influenced by several confounding factors. Predicting the effects of endodontic treatment in isolation remains challenging [31,32]. The local and systemic immunological effects of chronic apical periodontitis [33,34] and the impact of combined RCT and surgical treatment on serum CRP levels in populations with good general health have been previously reported [35,36]. The systemic effects may result from coexisting pulpal and chronic apical inflammation, triggered by anaerobic bacteria. This process, similar to plaque-induced periodontal disease, is associated with cytokine production and tissue destruction [37–39]. Furthermore, low-density lipoprotein (LDL) may interact with Toll-like receptor 4, similarly to how lipopolysaccharide (LPS) from the bacterial walls of periodontopathogenic bacteria functions, while Toll-like receptor 2 can also interact with LPS from Gram-positive bacteria and bind LDL [40]. This could represent an important pathway linking periodontal disease and hyperlipidaemia.

The systemic effects of concurrently treating pulpal and periapical inflammation in hyperlipidaemic patients have not been previously evaluated. Therefore, our interventional clinical study aimed to assess the impact of combined periodontal, endodontic, and dentoalveolar surgical treatments on serum CRP levels and serum lipid components and lipoproteins in patients with hyperlipidaemia. To the best of our knowledge, evidence on the effects of these treatment modalities on laboratory parameters in hyperlipidaemic patient groups is lacking. We hypothesised that in addition to the beneficial effects of treating periodontal disease, eliminating microbial sources associated with pulpal inflammation could provide adjunctive therapeutic support for hyperlipidaemic patients.

Methods:

-Participants:

Participants have been defined more in detail. We have added new text part.

Hyperlipidaemic patient group

Twenty-eight Caucasian patients with Fredrickson type IIa and IIb hyperlipidaemia were enrolled in the study. Type IIa, also known as familial hypercholesterolaemia, occurs due to a gene alteration and causes high levels of LDL cholesterol (LDL-C) (total cholesterol > 5.2 mmol/L, LDL-C > 3.0 mmol/L). Type IIb is a subvariation involving a gene alteration that causes both high cholesterol and high triglycerides (mixed hyperlipidaemia) (total cholesterol > 5.2 mmol/L, LDL-C > 3.0 mmol/L, triglycerides > 1.7 mmol/L) [41]. Lipid-lowering therapy included simvastatin (20–40 mg/day), atorvastatin (20–40 mg/day), fluvastatin (80 mg/day), or rosuvastatin (10–40 mg/day) as monotherapy or in combination with ezetimibe (10 mg/day) and/or fenofibrate (160 mg/day), administered according to current therapeutic guidelines.

  • The experimental and control groups must be matched in size.

Regarding the age of the hyperlipidemic patient group and within a 2 year time frame limited by financial factors only 18 controls could be enrolled in the study based on the strict inclusion and exclusion criteria That is could be a limitation of the study.

We added new limitation of the study section. Our study population was relatively small, due to the strict inclusion criteria, the need of rigorous, continuous control and well-balanced lipid-lowering therapy by specialists, the wide range of dental treatments, and the patient’s long-term compliance during the whole study period. Further investigations, involving a larger number of patients within the frame of a multi-center study may confirm and extend the findings of this study.

Limitation of the study

Our study population was relatively small, which can be attributed to the strict inclusion criteria, the requirement for rigorous and continuous specialist control of well-balanced lipid-lowering therapy, the wide range of dental treatments performed, and the need for long-term patient compliance throughout the study period. Further investigations involving a larger patient population, conducted within the framework of a multicentre study, may confirm and expand upon these findings.

  • The results should be abbreviated and presented in accordance with the CONSORT recommendations. Now it is very difficult to follow the results as the text is full of errors and written in uncorrected English.

Thank you for your comments! We added new reformulated text parts, Figures into the article.

  • As written, 15/28 patients in the test group were without periodontitis. The frequency of all dental pathologies should be reported, including the frequency and details of patients with mixed pathology. These data should be listed in the table.

Thank you for your comments! We added new reformulated text parts, Figures into the article.

Results

Based on the inclusion criteria, 38 hyperlipidaemic patients were initially enrolled in the study. Six participants declined to participate, and four did not complete the treatment period because of non-compliance. Ultimately, 28 patients with hyperlipidaemia (mean age, 53.36 ± 11.60 years; 11 men and 17 women) and 18 healthy controls (mean age, 51.22 ± 11.80 years; 6 men and 12 women) completed the study (Figure 1). During the study period, the participants reported no changes in lifestyle or medications, and no adverse events or side effects were observed.

Baseline characteristics

There were no significant differences in age between patients and healthy controls (p = 0.548). At baseline, the mean number of missing teeth was similar between the study group (10.32 ± 6.56) and healthy controls (7.39 ± 6.66) (p = 0.148). Significant differences (p < 0.01) were observed between patients and healthy controls in the PI (1.02 ± 0.37 vs. 0.34 ± 0.20), GI (0.71 ± 0.52 vs. 0.25 ± 0.25), and PPD (2.02 ± 0.54 vs. 1.50 ± 0.25 mm) (Table 1). Additionally, a significant difference was detected in body mass index (30.23 ± 6.40 vs. 26.06 ± 4.24 kg/m2) (p = 0.019; data not shown).

All 28 patients had chronic dental inflammation originating from pulpal or periodontal conditions. Of these, 5 patients were diagnosed with chronic gingivitis (according to EFP criteria, 2017 [42]), and 10 patients had a PPD of ≤3 mm. The remaining dental issues included pulp necrosis, chronic apical periodontitis, periapical cysts, insufficient RCT, radices, residual roots, and impacted teeth. Twelve patients had insufficient RCT, affecting 24 teeth, and in 9 cases, this was associated with chronic apical periodontitis. In three patients, chronic apical periodontitis was found in association with chronic pulpal inflammation, affecting three teeth. Four patients had residual roots (five roots in total), two of which were associated with chronic apical periodontitis. In one patient, a periapical cyst was identified and confirmed by postsurgical histology. The number of patients in each pathosis group is shown in Figure 3, while the number of teeth associated with the detected pathologies is displayed in Figure 4.

Figure 3. Number of patients categorised by pathosis groups

Figure 4. Number of teeth associated with the detected pathoses

Periodontal outcomes

Combined periodontal, endodontic, and dentoalveolar surgical treatments led to significant improvements in periodontal parameters 3 months after treatment. The PI decreased to 0.73 ± 0.47 (p < 0.01), the GI to 0.55 ± 0.39 (p < 0.05), and the PPD to 1.73 ± 0.42 mm (p < 0.01) (Table 1).

N

mean

[95% confidence interval]

SD

min

max

percentile

p*

25th

50th (median)

75th

(paired t-test/ independent t-test)

Age

baseline hyperlipidemic patients

28

53,3  [49,0-57,6]

11.60

32.00

71.00

43.50

54.00

63.00

(reference)

controls for CRP evaluation

28

52,1  [48,1-56,0]

10.69

32.00

71.00

44.50

54.50

58.75

0.677**

healthy control

18

51,2  [45,7-56,6]

11.80

32.00

74.00

41.50

50.50

61.25

0.548

Body Mass Index

baseline hyperlipidemic patients

28

30,2  [27,8-32,6]

6.40

18.97

43.75

25.66

29.95

35.18

(reference)

healthy control

18

26,0  [24,1-28,0]

4.24

20.00

36.14

23.86

25.15

27.64

0.019

Missing teeth

baseline hyperlipidemic patients

28

10,3  [7,89-12,7]

6.56

0.00

23.00

5.00

9.00

13.75

(reference)

healthy control

18

7,39  [4,31-10,4]

6.66

0.00

25.00

1.25

7.00

9.75

0.148

Periodontal status

PI

baseline hyperlipidemic patients

28

1,02  [0,89-1,16]

0.37

0.42

1.75

0.69

1.04

1.24

(reference)

3 months after treatment

28

0,73  [0,55-0,90]

0.47

0.00

1.80

0.31

0.70

1.03

<0.001

healthy control

18

0,34  [0,25-0,44]

0.20

0.13

0.91

0.20

0.31

0.43

<0.001

GI

baseline hyperlipidemic patients

28

0,71  [0,52-0,91]

0.52

0.10

1.67

0.26

0.59

1.11

(reference)

3 months after treatment

28

0,55  [0,40-0,69]

0.39

0.02

1.42

0.31

0.42

0.81

0.034

healthy control

18

0,25  [0,14-0,37]

0.25

0.02

0.90

0.07

0.20

0.31

<0.001

PPD

baseline hyperlipidemic patients

28

2,02  [1,82-2,22]

0.54

1.05

3.16

1.56

2.00

2.42

(reference)

3 months after treatment

28

1,73  [1,58-1,88]

0.42

1.10

2.73

1.39

1.71

2.04

0.001

healthy control

18

1,50  [1,39-1,62]

0.25

1.15

2.08

1.27

1.54

1.62

<0.001

Table 1. Clinical and periodontal characteristics in the control group and baseline versus 3-month results of hyperlipidaemic patients

*Differences between patients’ baseline and healthy controls’ periodontal status were analysed using independent t-tests. The treatment effect (baseline vs. 3-month results) for periodontal disease was evaluated using paired t-tests.

Significant differences were observed in the microbial analysis of subgingival and supragingival plaque samples obtained on admission between patients and healthy controls, as well as during the follow-up period within the patient group, with respect to the frequency of normal and pathogenic flora (p < 0.01). The detected species in dental plaque and sulcus samples are presented in Figures 5 and 6. The occurrence of pathogenic flora had decreased significantly by the 3-month follow-up investigation in both subgingival and supragingival plaque samples (p < 0.001) (Table 2).

Subgingival plaque samples on admission contained Prevotella species, including P. intermedia; Capnocytophaga species; Fusobacterium species; and Bacteroides species, such as B. eggerthii and B. caccae. Other detected organisms included Streptococcus intermedius, Gram-negative rods, Gram-negative cocci, and Gram-positive cocci. Notably, periodontal pathogens of the red complex were not cultivated. By the 3-month follow-up, normal flora was predominant in most samples, although Prevotella corporis and Gram-positive bacteria were still occasionally detected.

Supragingival plaque samples on admission contained Capnocytophaga species, Prevotella species (e.g., P. corporis, P. intermedia, P. oralis, P. disiens, P. loeschii), Bacteroides species, Fusobacterium, Gram-negative cocci and rods, and Gram-positive cocci. By the 3-month follow-up, a marked reduction in periodontopathogenic bacteria was observed. Capnocytophaga species, P. corporis, Enterobacter species, and Gram-negative and Gram-positive bacteria were detected less frequently and in smaller amounts.

Figure 5. Cultivable species in supragingival plaque samples according to the time of investigation

Figure 6. Cultivable species in sulcus samples according to the time of investigation

Type of examination

Type of the examined flora

p

normal (%)

pathogenic (%)

Supragingival plaque

baseline

15.38

84.62

(reference)

3 months after treatment

67.86

32.14

<0.001

healthy control

77.78

22.22

<0.001

Subgingival plaque

baseline

26.92

73.08

(reference)

3 months after treatment

92.86

7.14

<0.001

healthy control

83.33

16.67

<0.001

Table 2. Types of subgingival and supragingival flora at baseline and 3-month samples

Serum CRP and lipid levels

No significant difference in the serum CRP levels was observed between hyperlipidaemic patients and healthy controls at baseline (p = 0.14) (Table 3). However, 3 months after completing the periodontal, endodontic, and dentoalveolar surgical treatments, the serum CRP levels in the hyperlipidaemic patient group decreased significantly from 3.30 ± 2.94 to 2.29 ± 1.73 mg/L (p < 0.05) (Table 3). Regarding lipid levels, the applied lipid-lowering therapy resulted in no significant differences in LDL-C or HDL-C levels (p > 0.05), while a borderline difference was noted in total cholesterol levels (p = 0.051) and a significant difference in triglyceride levels between healthy controls and hyperlipidaemic patients at baseline (p < 0.05) (Table 3). A significant reduction in serum cholesterol levels was observed between the baseline and 3-month results (p < 0.05), with a borderline reduction in LDL-C (p = 0.057). No significant changes were found in triglyceride levels (p > 0.05), and HDL-C levels remained unchanged (p > 0.05) (Table 3.)

Spearman correlation analysis showed that the difference in the LDL-C level between baseline and 3 months was correlated with the difference in the GI between baseline and 3 months (R = 0.38, p = 0.04). However, repeated-measures analysis of variance revealed that changes in periodontal parameters during the 3-month follow-up period did not significantly affect changes in the measured serum parameters.

  • The frequency of all bacteria should be reported in a table. As this is not generally accepted, the distinction between pathogenic and normal species needs to be removed.

Unfortunately to the statistics we used pathogenic and normal flora as terms to differentiate the different flora.

  • The odds for lowering CRP as the primary outcome should be calculated in relation to the different terapeutic measures.

 Thank you for your suggestions, we added new discussion part into the text.

In this study, correlation analysis revealed that changes in the investigated parameters did not show simple associations but rather influenced each other in a complex manner. The patients included in this study did not have severe periodontitis, which is why characteristic values reflecting periodontal disease activity were analysed in relation to serum values. Although the potential systemic consequences of inflammatory lesions are not fully understood, their association with systemic parameters and general health has been previously described [66].

DISCUSSION

  • The first paragraph should summarise the most important results
  • The authors must place their results in the context of other, similar research data

 Thank you for your suggestions, we added new discussion part into the text.

Discussion

Our clinical study demonstrated that both non-surgical and surgical dental treatments led to improvements in clinical periodontal parameters (PI, GI, and PPD) and the microflora of dental plaque and sulcus samples, alongside reductions in the serum lipid and CRP levels. At the 3-month follow-up, significant decreases in the serum cholesterol and CRP levels were observed in the hyperlipidaemic patient group, and moderate improvement in LDL-C levels was also detected. Although the triglyceride and HDL-C levels improved, these changes were not statistically significant. The novelty of our protocol lies in its inclusion of not only periodontal but also pulpal-origin inflammation in the treatment regimen. This comprehensive approach to dental treatment in hyperlipidaemic patients has not been previously described, highlighting the potential systemic benefits of addressing multiple sources of dental inflammation.

At baseline, significant differences were observed in periodontal parameters (PI, GI, PPD) and microbial profiles of dental plaque and sulcus samples between the hyperlipidaemic patient group and the healthy control group, consistent with previous findings [10]. The potential associations and bidirectional relationships between hyperlipidaemia and dental infections such as periodontitis have been widely investigated [3,17–26]. The concept of overlapping and interconnected aetiopathological mechanisms in dental inflammatory diseases, which may increase the risk of hyperlipidaemia and thereby elevate the CVD risk, is well established [30,47]. Previous studies have shown that poor oral hygiene and hyperlipidaemia often co-exist [15,17,19,26]. Similarly, our findings align with studies evaluating the periodontal status of hyperlipidaemic patients [17,19–22,29], as supported by Fentoglu et al., with the hyperlipidaemic group exhibiting significantly higher periodontal parameters (GI, PI, PPD) than the healthy controls [17]. Despite lipid-lowering therapy in our study, no significant differences were observed in LDL-C or HDL-C levels, a borderline difference was found in total cholesterol levels, and a significant difference was noted in triglyceride levels between the groups at baseline, likely due to the lower efficacy of statin and fibrate therapy in reducing triglyceride levels [48–51]. Following causative therapy, including scaling and root planing, significant improvements were observed in all indices (PI, GI, and PPD) and in subgingival and supragingival plaque samples, aligning with previous findings [18]. Additionally, a notable shift in the microflora was observed, moving from a dominance of pathogenic bacteria toward a commensal microflora in both plaque and subgingival samples.

Odontogenic infection-related inflammatory signals and effector molecules contributing to dysregulated lipid metabolism have been previously reported [52,53], potentially explaining the observed improvements in serum metabolic parameters following intensive periodontal and endodontic treatments in healthy patients with periodontitis and apical periodontitis [54–56]. Earlier studies evaluated the effects of periodontal treatment on serum lipid parameters in both healthy and hyperlipidaemic patients. D’Aiuto reported that intensive periodontal therapy was more effective than standard treatments in improving serum lipid parameters [54]. A Turkish research group systematically studied the metabolic and inflammatory effects of lipid-lowering therapy combined with periodontal treatment in hyperlipidaemic patients [19,20,57]. In 2010, they demonstrated that a 3-month statin therapy significantly improved lipid profiles in hyperlipidaemic patients, and subsequent non-surgical periodontal treatment led to further reductions in serum total cholesterol and LDL-C levels. In our study, which incorporated combined non-surgical and surgical dental treatments, significant reductions in serum cholesterol levels and borderline reductions in LDL-C levels were observed, while HDL-C and triglyceride levels remained unchanged at the 3-month follow-up. The differing outcomes between Fentoglu et al. [19] and our study could be attributed to our protocol, which eliminated the short-term effects of lipid-lowering therapy by requiring a strict 6-month regimen of lipid-lowering medication, lifestyle modifications, and dietary adjustments under specialist supervision before initiating dental treatments, as opposed to their shorter, 3-month statin treatment prior to periodontal therapy.

Periodontal and pulpo-periapical infections are associated with systemic microinflammation [58,59]. Poornima et al. reported a significant decrease in CRP levels following RCT in healthy individuals [60]. In our study, serum CRP levels were assessed at baseline and at two follow-up points—1 week and 3 months after completing combined dental treatments. The 1-week follow-up allowed for monitoring rapid changes and immediate effects of the treatments on serum inflammatory parameters, while the 3-month follow-up evaluated their long-term impact [35]. Weight loss, diet, physical activity, and smoking cessation, as well as 6 weeks of statin or fibrate therapy, are known to reduce serum CRP levels [61–64]. To control these confounding factors, we implemented a strict protocol before initiating combined dental treatments. Our findings demonstrated a significant reduction in serum CRP levels at the 3-month checkpoint, consistent with Tawfig et al., who observed similar decreases in hyperlipidaemic patients with chronic periodontitis 3 months after non-surgical periodontal therapy [65]. Conversely, Fentoglu et al. reported no significant changes in CRP levels [20,22], and another study unexpectedly noted CRP elevation following statin therapy [19]. Apical periodontitis has previously been shown to elevate serum CRP levels by inducing low-grade systemic inflammation [66]. Treating apical periodontitis eliminates sources of toxins and bacterial by-products, potentially reducing immune system activation and mitigating systemic inflammation [60].

.

In this study, correlation analysis revealed that changes in the investigated parameters did not show simple associations but rather influenced each other in a complex manner. The patients included in this study did not have severe periodontitis, which is why characteristic values reflecting periodontal disease activity were analysed in relation to serum values. Although the potential systemic consequences of inflammatory lesions are not fully understood, their association with systemic parameters and general health has been previously described [66].

Limitation of the study

Our study population was relatively small, which can be attributed to the strict inclusion criteria, the requirement for rigorous and continuous specialist control of well-balanced lipid-lowering therapy, the wide range of dental treatments performed, and the need for long-term patient compliance throughout the study period. Further investigations involving a larger patient population, conducted within the framework of a multicentre study, may confirm and expand upon these findings.

Conclusion

Combined non-surgical and surgical dental treatments targeting local dental inflammations of pulpal and/or periodontal origin are associated with reductions in CRP and cholesterol levels. These findings suggest that such treatments may serve as a beneficial adjunct therapy alongside lipid-lowering therapy in patients with hyperlipidaemia.

Comments on the Quality of English Language

I suggest that you also correct the article linguistically, as it is difficult to read.

Thank you for your comment, we improved the english with a help of an english support.

Reviewer 3 Report

Comments and Suggestions for Authors

1. Study Population: The total number of included patients and controls should be explicitly stated in the abstract. This is essential for clarity and transparency.

2. Abstract Over-Interpretation: The conclusion in the abstract that "a decrease in cholesterol level due to dental care represents a smaller risk of cardiovascular disease" is an over-interpretation of the results. This assertion should be either removed or supported with more robust evidence.

3. Diagnostic Criteria for Hyperlipidemia: In line 173, the authors state, "Subjects with hyperlipidemia were selected from the subjects who went to the regular medical examination to the 1st Department of Internal Medicine, MHSC UD." However, the diagnostic criteria for hyperlipidemia are unclear. These criteria should be clearly described in the inclusion criteria, preferably earlier in the manuscript. Although partially mentioned in line 179, this information should be consolidated and moved earlier in the methods section for better clarity.

4. Sample Size Justification: The study included only 28 patients. Was this number sufficient to ensure the validity of the findings? The authors should provide a detailed explanation of how the sample size was determined, including power analysis or other statistical considerations.

5. Confounding Effect of Lipid-Lowering Therapy: The authors state that enrolled patients received lipid-lowering therapy. This is a major confounding factor, as any observed cholesterol-lowering effect after dental hygiene care could be attributed to the lipid-lowering therapy, not the dental care itself. The authors must address this issue and discuss how they controlled for it, if at all.

6. Control Group in Methods Section: The results indicate that there were 18 healthy controls. However, there is no mention of this control group in the methods section. The authors should clearly describe the process for recruiting and including healthy controls in the study design.

7. Homogeneity Between Study Groups: In line 266, the authors state that "There were no significant differences between patients and healthy controls by age (p=0.548)." Similarly, they mention that "There were no significant differences in the number of missing teeth between the study group (10.32±6.56) and healthy controls (7.39±6.66, p=0.148)."

However, the absence of statistically significant differences does not confirm that the two groups are homogenous in terms of age and missing teeth. If the authors aim to demonstrate homogeneity, they should apply more appropriate statistical methods, such as equivalence testing or other specific homogeneity assessments.

8. Statistical Methods and Results for Cholesterol: The absence of one-way ANOVA results for cholesterol in Table 3 is concerning. Instead, the authors conducted a Friedman test without explaining the rationale for this choice.

Moreover, the Friedman test results showed no significant differences among groups (p > 0.05), indicating that dental care did not significantly lower cholesterol levels in hyperlipidemia patients.

Despite this, the authors state that "There was a significant reduction in the serum cholesterol level between admission and 3 months after treatment results (p<0.05)." This appears to be based on a paired t-test between admission and 3-month results. However, using a paired t-test in this context is inappropriate and statistically incorrect given the nature of the data. This issue requires immediate correction and clarification.

Author Response

Reviewer 3

Open Review

We appreciate a lot the useful comments and suggestions to our manuscript entitled “Treatment of dental inflammatory lesions: effects on the laboratory parameters in patients with hyperlipidemia: a clinical interventional study (jcm-3352978)” by R. Martos et al. Below we would like to reply to the comments in a point-by-point fashion. Accordingly, we have introduced changes in the revised manuscript that are highlighted in yellow (in the revised manuscript). Some alterations in the revised manuscript represent a combination of the responses to more than one reviewer out of the three reviewers.

Comments and Suggestions for Authors

  1. Study Population: The total number of included patients and controls should be explicitly stated in the abstract. This is essential for clarity and transparency.

We added the total number of enrolled patients and controls into the abstract section.

Background: Patients with hyperlipidaemia are of interest because of the possible interplay between chronic local dental infections and hyperlipidaemia. This interventional clinical study aimed to evaluate the oral health status of hyperlipidaemic patients receiving lipid-lowering therapy for at least 6 months and the effects of non-surgical and surgical dental treatments on serum C-reactive protein (CRP) levels and lipid markers.

Methods: Twenty-eight patients with controlled hyperlipidaemia and 18 healthy controls were enrolled in the study. All participants underwent dental examinations (clinical evaluation, X-ray imaging, and microbial analysis of subgingival and supragingival plaque samples) at baseline. Hyperlipidaemic patients received periodontal, endodontic, and dentoalveolar surgical treatments. Serum CRP and lipid parameters were assessed at baseline, 1 week, and 3 months, while subgingival and supragingival plaque samples were analysed at baseline and 3 months after completing dental treatments.

Results: At the 3-month follow-up, clinical periodontal characteristics, including the plaque index, gingival index, and periodontal probing depth, improved significantly (p < 0.05). A significant shift in microflora was observed in both subgingival and supragingival plaque samples (p < 0.05), alongside improvements in periodontal values and a significant reduction in serum CRP levels (p < 0.05). Serum cholesterol levels decreased significantly, while moderate improvements in serum triglycerides, low-density lipoprotein, and high-density lipoprotein levels were observed but were not statistically significant (p > 0.05).

Conclusions: Treating local dental inflammation is associated with a significant decrease in CRP and cholesterol levels and may serve as a beneficial adjunct therapy alongside lipid-lowering therapy in patients with hyperlipidaemia.

  1. Abstract Over-Interpretation: The conclusion in the abstract that "a decrease in cholesterol level due to dental care represents a smaller risk of cardiovascular disease" is an over-interpretation of the results. This assertion should be either removed or supported with more robust evidence.

 Thank you for your suggestions, we added new abstract section (see above):

  1. Diagnostic Criteria for Hyperlipidemia: In line 173, the authors state, "Subjects with hyperlipidemia were selected from the subjects who went to the regular medical examination to the 1st Department of Internal Medicine, MHSC UD." However, the diagnostic criteria for hyperlipidemia are unclear. These criteria should be clearly described in the inclusion criteria, preferably earlier in the manuscript. Although partially mentioned in line 179, this information should be consolidated and moved earlier in the methods section for better clarity.

 Thank you for your suggestions. The diagnostic criteria of hyperlipidemia were the following:

Twenty-eight Caucasian patients with Fredrickson type IIa and IIb hyperlipidaemia were enrolled in the study. Type IIa, also known as familial hypercholesterolaemia, occurs due to a gene alteration and causes high levels of LDL cholesterol (LDL-C) (total cholesterol > 5.2 mmol/L, LDL-C > 3.0 mmol/L). Type IIb is a subvariation involving a gene alteration that causes both high cholesterol and high triglycerides (mixed hyperlipidaemia) (total cholesterol > 5.2 mmol/L, LDL-C > 3.0 mmol/L, triglycerides > 1.7 mmol/L) [41]. Lipid-lowering therapy included simvastatin (20–40 mg/day), atorvastatin (20–40 mg/day), fluvastatin (80 mg/day), or rosuvastatin (10–40 mg/day) as monotherapy or in combination with ezetimibe (10 mg/day) and/or fenofibrate (160 mg/day), administered according to current therapeutic guidelines (D S Fredrickson, R S Lees A System for Phenotyping Hyperlipoproteinemia Circulation. 1965 Mar:31:321-7. doi: 10.1161/01.cir.31.3.321.PMID: 14262568) We added this section into Materials and Method section, Participant subsection.

  1. Sample Size Justification: The study included only 28 patients. Was this number sufficient to ensure the validity of the findings? The authors should provide a detailed explanation of how the sample size was determined, including power analysis or other statistical considerations.

 Thank you for your suggestions. The required sample size was calculated based on the standard deviation of the CRP levels (1.3 mmol/L) among patients without malignant or inflammatory diseases, as estimated by the Department of Laboratory Medicine, with a 1-mmol/L CRP concentration defined as the minimum detectable change. Age- and sex-matched healthy controls (n = 28; mean age, 52.11 ± 10.69 years; 10 men and 18 women) for CRP evaluation were selected from the control pool of the TECH-09-A1-2009-0113 (mAB-CHIC) project of the Department of Laboratory Medicine, MHSC UD. Assuming equal group sizes for controls and patients, a 5% type I error, and 80% statistical power, the required minimum sample size was 27 patients in the diseased group. The CRP control group data was used exclusively for sample size determination.

  1. Confounding Effect of Lipid-Lowering Therapy: The authors state that enrolled patients received lipid-lowering therapy. This is a major confounding factor, as any observed cholesterol-lowering effect after dental hygiene care could be attributed to the lipid-lowering therapy, not the dental care itself. The authors must address this issue and discuss how they controlled for it, if at all.

 Thank you for your suggestions. The administered lipid lowering therapy was the following :Simvastatin (20-40 mg/d), atorvastatin (20-40 mg/d), fluvastatin (80 mg/d) or rosuvastatin (10-40 mg/d) monotherapy or in combination with ezetimibe (10 mg/d) and/or fenofibrate (160 mg/d) was administered according to the current therapeutic guidelines.

The main goal of the delayed enrollment is to exclude the possible confounding effect of lipid-lowering therapy. Therefore, the types and doses of the lipid-lowering agents were kept constant throughout the study. To eliminate the short-term effects of lipid-lowering therapy, we enrolled patients with hyperlipidemia after the adjustment of the combined dietary and statin/fibrate therapy for at least six months. Therefore, the confounding effect of these agents on lipid and inflammatory parameters could be excluded. In the patient group, we detected similar levels of lipids throughout the study. The stable serum lipid levels proved the effective control of hyperlipidemia and compliance with medication.

We added this text part also into the Materials and method section, Participant subsection.

  1. Control Group in Methods Section: The results indicate that there were 18 healthy controls. However, there is no mention of this control group in the methods section. The authors should clearly describe the process for recruiting and including healthy controls in the study design.

 Thank you for your suggestions. Control group

Age- and sex-matched healthy controls willing to participate in this study were selected from patients attending check-ups at the Department of Restorative Dentistry, Faculty of Dentistry, MHSC UD. Controls were selected to exclude individuals with hyperlipidaemia or hypertension, and smokers were also excluded. Body mass index was calculated by a physician. . The lifestyle of participants, including exercise, diet, and daily brushing habits were self-reported. Body mass index (BMI) was calculated. unfortunately within a 2 year time frame limited by financial factors only 18 controls could be enrolled in the study based on the strict inclusion and exclusion criteria. That is could be a limitation of the study.

  1. Homogeneity Between Study Groups: In line 266, the authors state that "There were no significant differences between patients and healthy controls by age (p=0.548)." Similarly, they mention that "There were no significant differences in the number of missing teeth between the study group (10.32±6.56) and healthy controls (7.39±6.66, p=0.148)."

 Thank you for your suggestions. The word homogeneity does not even appear in the text. The tests cited by the reviewer simply show that there were no significant differences between the groups.

However, the absence of statistically significant differences does not confirm that the two groups are homogenous in terms of age and missing teeth. If the authors aim to demonstrate homogeneity, they should apply more appropriate statistical methods, such as equivalence testing or other specific homogeneity assessments.

  1. Statistical Methods and Results for Cholesterol: The absence of one-way ANOVA results for cholesterol in Table 3 is concerning. Instead, the authors conducted a Friedman test without explaining the rationale for this choice.

Moreover, the Friedman test results showed no significant differences among groups (p > 0.05), indicating that dental care did not significantly lower cholesterol levels in hyperlipidemia patients.

 Thank you for your suggestions. ANOVA is a parametric test, the nonparametric version of which is the Friedman test. We used the Friedman test because the data were not normally distributed.

Despite this, the authors state that "There was a significant reduction in the serum cholesterol level between admission and 3 months after treatment results (p<0.05)." This appears to be based on a paired t-test between admission and 3-month results. However, using a paired t-test in this context is inappropriate and statistically incorrect given the nature of the data. This issue requires immediate correction and clarification.

 Thank you for your suggestions. Cholesterol: As shown in Table 3, cholesterol levels decreased significantly 3 months after admission; since the data were not normally distributed, a parametric test (e.g., t test) could not be used; the corresponding p-value is in the last column because a non-parametric test (Wilcoxon test) was used.

Thank you for your comments. According to your suggestions we added new text parts (highlighted in yellow in the manuscript)

Spearman correlation analysis showed that the difference in the LDL-C level between baseline and 3 months was correlated with the difference in the GI between baseline and 3 months (R = 0.38, p = 0.04). However, repeated-measures analysis of variance revealed that changes in periodontal parameters during the 3-month follow-up period did not significantly affect changes in the measured serum parameters.

In this study, correlation analysis revealed that changes in the investigated parameters did not show simple associations but rather influenced each other in a complex manner. The patients included in this study did not have severe periodontitis, which is why characteristic values reflecting periodontal disease activity were analysed in relation to serum values. Although the potential systemic consequences of inflammatory lesions are not fully understood, their association with systemic parameters and general health has been previously described [66].

Limitation of the study

Our study population was relatively small, which can be attributed to the strict inclusion criteria, the requirement for rigorous and continuous specialist control of well-balanced lipid-lowering therapy, the wide range of dental treatments performed, and the need for long-term patient compliance throughout the study period. Further investigations involving a larger patient population, conducted within the framework of a multicentre study, may confirm and expand upon these findings.

Conclusion

Combined non-surgical and surgical dental treatments targeting local dental inflammations of pulpal and/or periodontal origin are associated with reductions in CRP and cholesterol levels. These findings suggest that such treatments may serve as a beneficial adjunct therapy alongside lipid-lowering therapy in patients with hyperlipidaemia.

Round 2

Reviewer 1 Report

Comments and Suggestions for Authors

The authors made the suggested modifications, significantly improving the quality of the manuscript. However, I believe the article's title still does not adequately reflect the study's objective, which is to evaluate these parameters during dental treatment. Please, revise the article's title.

Author Response

The authors made the suggested modifications, significantly improving the quality of the manuscript. However, I believe the article's title still does not adequately reflect the study's objective, which is to evaluate these parameters during dental treatment. Please, revise the article's title.

Thank you for your valuable contribution to our work. The new title is:

Do have any beneficial adjunctive effects of combined periodontal, endodontic and dentoalveolar surgical treatments on laboratory parameters in patients with hyperlipidaemia? a clinical interventional study

Reviewer 3 Report

Comments and Suggestions for Authors

Authors addressed all issues raised by reviewer. I don't have further comment on this article.

Author Response

Suggestions for Authors

Authors addressed all issues raised by reviewer. I don't have further comment on this article.

Thank you for your valuable contribution to our work.